# Responses of Microbial Metabolic Rates to Non-Equilibrated Silicate versus Calcium-Based Ocean Alkalinity Enhancement

Laura Marín-Samper[1], Javier Arístegui[1], Nauzet Hernández-Hernández[1], Ulf Riebesell[2]

[1] Instituto de Oceanografía y Cambio Global, Universidad de Las Palmas de Gran Canaria, 35017 Telde, Spain
[2] GEOMAR Helmholtz Centre for Ocean Research Kiel, 24148 Kiel, Germany

*Correspondence to*: Laura Marín-Samper (laura.marin@ulpgc.es), Javier Arístegui (javier.aristegui@ulpgc.es)

**Abstract.** This study contributes to the inaugural exploration of non-equilibrated ocean alkalinity enhancement (OAE). Total alkalinity (TA) was manipulated, with silicate and calcium-based $\Delta$TA gradients ranging from 0 to 600 µmol · L$^{-1}$, without prior $CO_2$ sequestration, under natural conditions and at a mesocosm scale (~60 m$^3$). This manipulation led to a sustained increase in pH and a decrease in $pCO_2$ throughout the experiment, as full natural equilibration through sea-air gas exchange did not occur. Implemented in a neritic system under post-bloom conditions, a midway mixing event was simulated. After the inorganic nutrient addition, mild delays in bloom formation were observed. These delays were related, though not directly proportional to, the $\Delta$TA gradient, as indicated by the gross production (GP), net community production (NCP), and the chlorophyll *a* (Chl*a*) concentrations. Notably, the delay was more pronounced for the calcium treatment set compared to the silicate one, with the low TA treatments exhibiting earlier responses than the high TA ones. This delay is likely due to the previously documented species-specific negative relationships between high pH/low $pCO_2$ conditions and phytoplankton growth rates. This study underscores the need for further investigation into the implications of these response patterns in terms of trophic transfer and seasonal suitability. Moreover, it is anticipated that a greater delay in bloom formation would be evident with a larger non-equilibrated TA gradient, highlighting the importance of exploring variations in TA thresholds for a comprehensive understanding of the OAE's impacts.

**Keywords.** OAE, alkalinization, silicate-based, calcium-based, community production, metabolic rates

## 1. INTRODUCTION

To reduce the concentration of atmospheric carbon dioxide ($CO_2$), and to be able to stay below the 1.5 to 2 ºC global mean temperature increase relative to preindustrial times, the realistic emissions cut alone is projected not to be sufficient (Friedlingstein et al., 2022, 2019; Fuss et al., 2020; Lee et al., 2021). The changes to our technological and socio-economic systems that would be necessary to attain the required emissions reductions can take decades, or longer, to be

implemented (Renforth and Henderson, 2017). In fact, all the projections that simulate the agreed upon conditional and unconditional Nationally Determined Contributions (NDCs) in terms of emissions cuts, and that also assume gross negative emissions, still fall well above those in which the temperature is restricted to the targeted maximum increase (IPCC, 2023). The latter require more extensive emissions reductions, alongside reaching net-zero emissions by 2050, and net negative emissions for the rest of the century (O'Neill et al., 2016; Rogelj et al., 2018). Therefore, the need for atmospheric carbon removal and sequestration is imperative to avoid the serious, long-term climatic consequences associated with surpassing the aforementioned temperature limit, which is considered a tipping point.

Ocean alkalinity enhancement (OAE) is a marine carbon dioxide removal (mCDR) approach that shows great promise. Model studies indicate that OAE shows the potential to remove atmospheric carbon on a gigaton (Gt) scale (Feng et al., 2017; Harvey, 2008), although it not only shows potential for carbon capture and long-term sequestration. It is also known to possibly aid in the allayment of ocean acidification (OA; Albright et al., 2016; Gattuso et al., 2018; Feng et al., 2017; Harvey, 2008). OAE is attained through the addition, in various ways, of alkali or alkaline compounds to seawater increasing total alkalinity (TA), pushing the carbonate equilibrium system from $CO_2$ to the bicarbonate ($HCO_3^-$) and carbonate ($CO_3^{-2}$) species (Kheshgi, 1995). This process allows for additional $CO_2$ diffusion in the course of regaining balance with the atmosphere and alleviates the effects of OA by increasing the ocean's buffering capacity.

$CO_2$ equilibration can be induced prior to the alkalinity addition, or the alkalinity plume can be left to equilibrate naturally through sea-gas exchange, which is more feasible. Indeed, large scale equilibrated OAE application would require the use of reactors to $CO_2$ equilibrate the alkaline solutions before deployment (Hartmann et al., 2023). If left to natural sea-gas exchange, however, this process can take several months to years (Jones et al., 2014). Notably, when carbon dioxide is not chemically pre-sequestered, the alterations to the carbonate system become more prominent. These consist of a substantial decrease in $CO_2$ partial pressure ($p CO_2$) and a subsequent significant increase in pH, particularly when compared to methods involving pre-equilibration. Such reduced $p CO_2$ resulting from the alkalinity manipulation without prior equilibration could potentially lead to $CO_2$ limitation among phytoplankton (Riebesell et al., 1993).

Past studies have in fact reported taxon-specific responses in phytoplankton growth based on the combined effects of $p CO_2$ and $H^+$ concentration in the context of OA (e.g., Paul & Bach, 2020). Before OA became a central focus of scientific research, high pH/low $CO_2$ conditions were observed to cause declines in marine phytoplankton growth rates (Goldman, 1999; Hansen, 2002). Notably, under air-equilibrated conditions, species-specific half saturation values ($k_{1/2}$) for $HCO_3^-$ and $CO_2$ acquisition in photosynthesis have also been reported (Raven and Johnston, 1991). Hence, given that the utilization of non-equilibrated OAE will entail significant changes to these

carbonate system parameters ($pCO_2$ and $H^+$), a response in terms of microbial production rates in relation to the deployed non-equilibrated alkalinity gradient was expected.

Another factor of uncertainty when considering OAE implementation is the source mineral type, whether it is calcium or silicate based. For silicate-based OAE, naturally occurring olivine-rich minerals, such as dunite, are being considered (Montserrat et al., 2017; Renforth and Henderson, 2017). This is because olivine occurs commonly in nature and weathers relatively quickly, eliminating the need for energy-intensive chemical processing prior to dissolution (Renforth and

Henderson, 2017; Schuiling and Krijgsman, 2006). However, olivine is comprised of forsterite ($Mg_2SiO_4$) and fayalite ($Fe_2SiO_4$) in a 9:1 ratio. An iron (Fe) addition may have a fertilizing effect on phytoplankton in the photic zone (Bach et al., 2019a; Hauck et al., 2016; Renforth and Henderson, 2017), and it is the Mg end member of olivine that, as it weathers, consumes atmospheric $CO_2$ naturally (Köhler et al., 2013; Renforth and Henderson, 2017). Additionally,

olivine dissolution releases silicon, which may benefit silicifying plankton species (Bach et al., 2019a; Hauck et al., 2016), but also other potentially harmful by-products such as nickel (Xin et al., 2023), and other trace metals (Bach et al., 2019a).

A calcium-based mineral that is being considered for OAE deployment is hydrated lime ($Ca(OH)_2$; Kheshgi, 1995). It is produced through the calcination of limestone and would dissolve

much faster than any natural mineral (Renforth and Henderson, 2017). Besides, the latter does not contain any dissolution by-products that could in theory negatively impact biota. However, the introduction of calcium to the system may promote calcification (Albright et al., 2016; Bach et al., 2019a), a process through which $CO_2$ is released (Zeebe and Wolf-Gladrow, 2001). Thus, that would entail a reduction in the OAE's carbon capture efficiency, alongside benefiting benthic

(Albright et al., 2016) and potentially pelagic calcifiers, so possibly inducing a plankton community composition shift away from silicifiers (Bach et al., 2019a).

Considering all these unknowns, the current study aimed at monitoring the response of the microbial community to a silicate versus a calcium-based, non-$CO_2$ equilibrated OAE implementation. To isolate the effects of the carbonate chemistry alterations in both case

scenarios, and to avoid confounding variables due to impurities in the raw minerals, simulations of a forsterite ($Mg_2SiO_4$) and a hydrated lime ($Ca(OH)_2$) addition were employed. Compounds that contain the key elements present in them separately, already in solution, were used. Gross and net oxygen production rates (GP and NCP respectively), community respiration (CR) rates, metabolic balance (GP:CR), and chlorophyll *a* (Chl*a*) concentration were monitored over a 53-

105   day period. Alterations driven by the mineral type, and the carbonate chemistry changes to these metabolic rates were expected (Bach et al., 2019a; Paul and Bach, 2020; Riebesell et al., 1993). Addressing the knowledge gaps explained above is key in understanding the non-equilibrated

OAE's environmental impacts, its potential for $CO_2$ removal in terms of efficiency and long-term sequestration, and thus in choosing a suitable approach for its safe deployment.

## 2. MATERIALS AND METHODS

### 2.1 Experimental setup and sampling

The experiment (KOSMOS Bergen 2022) was carried out in Raunefjorden, 1.5 km offshore from the Espegrend Marine Research Field Station, of the University of Bergen, Norway, under post-bloom conditions, starting on the 7th of May 2022. This location provided protection from swells and access to all the necessary facilities to conduct the experiment right onshore (Ferderer et al., 2023). Ten KOSMOS (Kiel Offshore Mesocosms for Ocean Simulations; Riebesell et al., 2013) units, or mesocosms, of an approximately 60 $m^3$ capacity were deployed. The mesocosm cylindrical bags (20 m long and 2.5 m in diameter) were left submerged and opened at the bottom for approximately a week prior to the start of the experiment on the 13th of May. Therefore, allowing for enough open water exchange to enclose a natural planktonic community as homogeneously as possible amongst all mesocosms. To close them, the sediment traps (2 m long, funnel shaped) were placed on the bottom ends, the tops of the bags were drawn out 1m off the water's surface, and the mesocosm roofs were put in place. In order to exclude all unevenly allocated large organisms, a ring with the same diameter as the mesocosms (Riebesell et al., 2013), and with a 1 mm net attached to it, was pulled from bottom to top, just after mesocosm closure (Day 0). Two days after closure, the volume contained in the mesocosms was determined. To do that, the water inside the mesocosms was first homogenized using a spider-like dispensing device (named henceforth "spider", see Riebesell et al., 2013) to bubble compressed air up and down the water column. Later, by measuring salinity before and after adding 50 L of a precisely calibrated NaCl brine solution, also using the spider, the volume contained in each mesocosm could be calculated, as in Czerny et al. (2013). Mesocosm inside and outside cleaning was carried out once a week. Outside cleaning was done by divers and people assisting from boats, using brushes. The inside cleaning was realized sinking a ring with the same diameter as the mesocosms, with rubber blades around its circumference, and pulling it up. This way, removing any growth found in the mesocosm walls that could interfere with the results of the experiment.

Samples from the entire 20 m water column inside the mesocosms, and from the fjord, were collected every two days using Integrated Water Samplers (IWS III, HYDRO-BIOS Apparatebau GmbH, Altenholz, Germany) with a 5 L internal volume capacity. Mesocosms were sampled in a random order and an extra sample from the fjord was always taken from the same location right next to mesocosms 5. After three sampling days (day 6), to properly monitor the starting conditions, the TA manipulation was applied. The experiment ended on the 6th of July, lasting 53

days. For further information on all the research and maintenance activities carried out throughout the experimental period, please refer to Supp. Fig. S1.

## 2.2 Carbonate chemistry manipulation and nutrient fertilization

The mesocosms were divided into two sets of five (see Table 1). Five mesocosms were treated with $CaCl_2 \cdot H_2O$, and with NaOH (Merck) setting a total alkalinity (TA) gradient from $\Delta TA$ 0, in increments of 150 $\mu mol \cdot L^{-1}$, up to a $\Delta TA$ of 600 $\mu mol \cdot L^{-1}$. The other set of five were treated with $MgCl_2 \cdot 6\ H_2O$, $Na_2SiO_3 \cdot 5\ H_2O$, and with NaOH (Merck) following the same TA gradient. The amounts of $Mg^{2+}$ and $Ca^{2+}$ were increased in proportion to the NaOH addition, whereas the amount of $Na_2SiO_3$ added was the same (75 $\mu mol \cdot L^{-1}$ target concentration) in all the silicate based OAE treatments, including the control. The TA increase of 2:1 with respect to the amount of $Na_2SiO_3$ added was taken into consideration by reducing the amount of NaOH accordingly, and by adding diluted HCl in the silicate-based control (Ferderer, et al., 2023). The target amounts of NaOH, $Na_2SiO_3$ and $MgCl_2$, and $CaCl_2$ for each treatment were dissolved in 20 L of MiliQ water. These were later added to the mesocosms, evenly up and down the water column, using the spider.

An inorganic nutrient addition was carried out on day 26 due to the oligotrophic conditions found inside the mesocosms compared to those observed in the fjord. But also, because the community inside was close to reaching heterotrophic balance even though the light conditions were reasonably optimal. The nitrate ($NO_3^-$) target concentration was set at 4 $\mu mol\ L^{-1}$ to simulate a mixing event that would promote a phytoplankton boom comparable in biomass to natural occurrences in the area. Phosphate ($PO_4^{3-}$) was added following the N:P Redfield ratio of 16:1 across all mesocosms, and silicate was added in an Si:N ratio of 1:4 only to the calcium treatments. This ratio was chosen to provide a niche for coccolithophores, preventing them from being outcompeted by diatoms, which would be silicate limited under these conditions (Gilpin et al., 2004; Schulz et al., 2017). Further, this design choice ensured that the Si content in the Ca treatments remained reasonably low, preserving the integrity of the mineral treatment differentiation, plus it roughly mirrored the silicate concentration in the fjord at the time. A second addition on day 28 was undertaken to correct for stoichiometric differences between mesocosms (see Ferderer, et al., 2023).

## 2.2 Carbonate chemistry

Samples for TA, dissolved inorganic carbon (DIC), and total seawater pH were collected into 250 mL glass flasks, allowing for plenty overflow, directly from the IWSs. TA and pH samples were collected every two days, whereas DIC was only measured on day 9. Samples were later sterile filtered (25mm, 0.2 $\mu m$ PES membrane, syringe filters; Filtropur S, SARSTEDT, Nümbrecht, Germany) with a peristaltic pump and with special care to avoid sea-gas exchange. TA concentrations were determined by a potentiometric two step titration using a Metrohm 862

Compact Titrosampler with HCl 0.05 M as the titrant, an Aquatrode Plus (Pt1000), and a 907 Titrando unit, as in Chen et al., 2022. DIC concentrations were measured with an AIRICA system (Marianda, Kiel, Germany; see Gafar & Schulz, 2018, and Taucher et al., 2017) with a differential gas analyzer (LI-7000, LI-COR Biosciences GmbH, Bad Homburg, Germany) at room temperature and within 12 h. Also, total seawater pH samples were acclimated to 25ºC in a thermostatic bath and later measured spectrophotometrically, at the same temperature, with a VARIAN Cary 100 in a 10 cm cuvette (Dickson et al., 2007). The rest of the carbonate system parameters were calculated with CO2Sys v2.5 (Lewis and Wallace, 1998) using the measured TA, total seawater pH, and nutrient concentrations, as well as the in-situ temperature and salinity daily means obtained from the CTD casts (see Sup. Fig. S1). pH was corrected against the measured DIC on day 9 (see Schulz et al., 2023).

### 2.3 Dissolved inorganic nutrient concentrations

Triplicate samples to account for technical variability were collected every two days. These were sterile filtered using 33mm, 0.45 µm PES membrane syringe filters (Filtropur S, SARSTEDT, Nümbrecht, Germany) within one to two hours after collection, and kept in the dark and at ambient temperature until analysis less than 6 hours later. Nitrate ($NO_3^-$), nitrite ($NO_2^-$), phosphate ($PO_4^{3-}$), and silicate ($Si(OH)_4$) concentrations were determined spectrophotometrically as in Hansen & Koroleff (1999). Ammonia ($NH_4^+$) was measured using a 10-AU Fluorometer (TURNER designs, San Jose, CA, USA) following Holmes et al. (1999).

### 2.4 Metabolic rates through oxygen production and consumption

Gross production (GP), net community production (NCP), and community respiration (CR) rates were determined by oxygen production and consumption in calibrated 125 mL nominal volume soda lime glass bottles following the Winkler method and the recommendations from Carpenter (1966), Bryan et al. (1976), and Grasshof et al. (1999), also described by Marín-Samper et al. (2024). Polycarbonate bottles were filled with 4.5 L of seawater per mesocosm, and from the fjord on each sampling day directly from the IWSs and brought to the lab. Out of each 4.5 L sample, ten soda lime bottles were first rinsed with sample water and then randomly filled, allowing ample overflow, using a silicone tube with an attached 280 µm mesh on one end. Four out of the ten subsamples per mesocosm were fixed right after collection, "initials", through the addition of 1 mL of a manganese chloride ($MnCl_2$) solution, and 1 mL of a sodium iodide (NaI) based alkaline solution, in this order. They were later covered with an opaque piece of fabric and stored upright in a rack underwater. Another three bottles were incubated inside opaque bags, namely "dark" ones, and the remaining three were incubated under "light" conditions. The latter were randomly distributed inside clear methacrylate incubators set outside, which were covered with a blue foil (172 Lagoon Blue foil, Lee filters, Burbank, USA) to better simulate the light spectrum of the

water column, along with the "initials" (already fixed and left upright to allow for the Mn(OH)$_2$ precipitate to settle), and the bags containing the "dark" ones. The incubators were hooked to a constant water flow system that siphoned water directly out of the fjord (from 14 m depth), into the incubator, and out into the fjord again. Data loggers (HOBO UA-002-64, Australia/New Zealand) were placed inside the incubators to monitor the temperature (roughly 11.11 and 10.34 °C in average during the day and night, respectively) and light (ranging from 0.20 to 688.89 µmol photons m$^{-2}$ s$^{-1}$) conditions throughout the experiment. After an incubation period of 24 hours, all samples were fixed and left to sediment for at least 2 hours. Finally, samples were acidified with 1 mL of 5 M sulphuric acid (H$_2$SO$_4$) and analysed with an automated titration system, with colorimetric end-point detection (Dissolved Oxygen Analyzer, SIS Schwentinental, Germany), using a 0.25 M sodium thiosulphate solution (Na$_2$S$_2$O$_3$ * 5H$_2$O) as the titrant. The mean of each set of replicates was used to calculate CR, NCP, and GP rates, using the following Eq. (1), Eq. (2) and Eq. (3) respectively:

$$CR \ [\mu mol \ L^{-1} h^{-1}] = \frac{Conc_I - Conc_D}{h_D} \tag{1}$$

$$NCP \ [\mu mol \ L^{-1} h^{-1}] = \frac{Conc_L - Conc_I}{h_L} \tag{2}$$

$$GP \ [\mu mol \ L^{-1} h^{-1}] = CR + NCP \tag{3}$$

where $Conc_I$, $Conc_D$ and $Conc_L$ correspond to the mean oxygen concentration of the initial, dark, and light samples, respectively, and $h_L$ and $h_D$ stand for the light and dark samples' incubation time in hours. The metabolic balance was later calculated by dividing the obtained GP rates by the CR rates.

Due to a COVID outbreak in the base, the two scientists in charge of measuring this parameter had to be confined. Therefore, data were not collected on days 47 and 49.

## 2.5 Chlorophyll *a* concentration

Chlorophyll *a* (Chl*a*) samples for each mesocosm were collected into 500 - 1000 mL dark bottles from 10L canisters that were filled directly from the IWSs. Samples were filtered onto glass fiber filters (Whatman grade GF/F, 0.7 µm nominal pore size), with minimal light exposure. Filters were stored in plastic vials at -80 ºC and analyzed fluorometrically after Welschmeyer (1994) the following day.

## 2.7 Data analyses

Data for days 47 and 49 for GP were calculated using the equation from a Spearman correlation model with Chl*a*, with both variables transformed to base-10 logarithm (Supp. Fig. S2). NCP rates for those two days were calculated using the equation obtained through also a Spearman correlation model of NCP with GP (Supp. Fig. S2). The calculated NCP was then subtracted from

the estimated GP to obtain the CR values for those two days. Daily linear regressions in relation to the target $\Delta$TA were carried out to determine the evolution of the TA effect on the GP and CR rates, on the Chl*a* concentration, on the metabolic balance (GP:CR) and on the assimilation numbers (GP:Chl*a*). To aid in the system's response interpretation, the experiment was divided into two phases: pre-nutrient addition phase (phase I: from day 7 to day 25), and a post-nutrient addition phase (phase II: from day 27 to day 53).

## 3. RESULTS

### 3.1 Carbonate chemistry and inorganic nutrient concentrations

There were minor discrepancies between the target TA concentrations and those actually achieved. As the gradient increased, the differences between the expected and the attained TA concentrations widened. (Table 1). Additionally, no significant differences were observed between the meant to be equivalent, calcium and silicate-based treatments in TA, and thus in $p$CO$_2$ and pH either (Figure 1).

**Table 1. The targeted and established carbonate chemistry treatments are compared here. The first column corresponds to the mesocosms (MK), and next to it the source mineral type. The target $\Delta$total alkalinity (TA) gradient, in $\mu$mol $\cdot$ L$^{-1}$, is listed for each MK, with the measured or "attained" mean $\Delta$TA, and finally the difference between the theoretical/target gradient and the actual mean $\Delta$TA obtained.**

| MK | Mineral | Target $\Delta$TA | Attained $\Delta$TA | Target-Attained |
|----|---------|-------------------|---------------------|-----------------|
| M5 | Ca | 0 | 0 | 0 |
| M1 | Ca | 150 | 138.4 | 11.6 |
| M9 | Ca | 300 | 287.5 | 12.5 |
| M7 | Ca | 450 | 404.0 | 46.0 |
| M3 | Ca | 600 | 530.0 | 70.0 |
| M6 | Si | 0 | 0 | 0 |
| M10 | Si | 150 | 140.3 | 9.7 |
| M2 | Si | 300 | 278.9 | 21.1 |
| M4 | Si | 450 | 422.4 | 27.6 |
| M8 | Si | 600 | 550.4 | 49.6 |

The mean pH levels ranged from $8.06 \pm 0.02$ to $8.71 \pm 0.03$, and from $8.03 \pm 0.02$ to $8.72 \pm 0.02$, in the calcium and silicate treatments respectively (Figure 1 B). In terms of mean $p$CO$_2$, the difference between the controls and the highest treatments was substantial, going from $372.3 \pm 19.7$ and $397.05 \pm 20.4$ $\mu$atm, in the calcium and silicate controls, down to $76.08 \pm 7.4$ and $72.6 \pm 5.07$ $\mu$atm in the highest treatments, correspondingly (Figure 1 C).

A noteworthy result is that full natural equilibration did not occur after 49 days. CO$_2$ incursion was minor, as it is clear from the relatively stable pH and $p$CO$_2$ levels observed in the treated mesocosms during the whole experiment (Figure 1 B and C). In fact, when deducting the

calculated $pCO_2$ levels obtained on day 53, to those observed right after the TA manipulation on day 7, $pCO_2$ increased between 63.9 and 12.75 µatm in the treated mesocosm. The gap in absolute

$pCO_2$ values being higher in the low $\Delta$TA treatments and declining towards high $\Delta$TA ones (Supp. Table S1). Therefore, neither the pH nor the $pCO_2$ in the mesocosms where TA was manipulated reached ambient levels throughout the experiment.

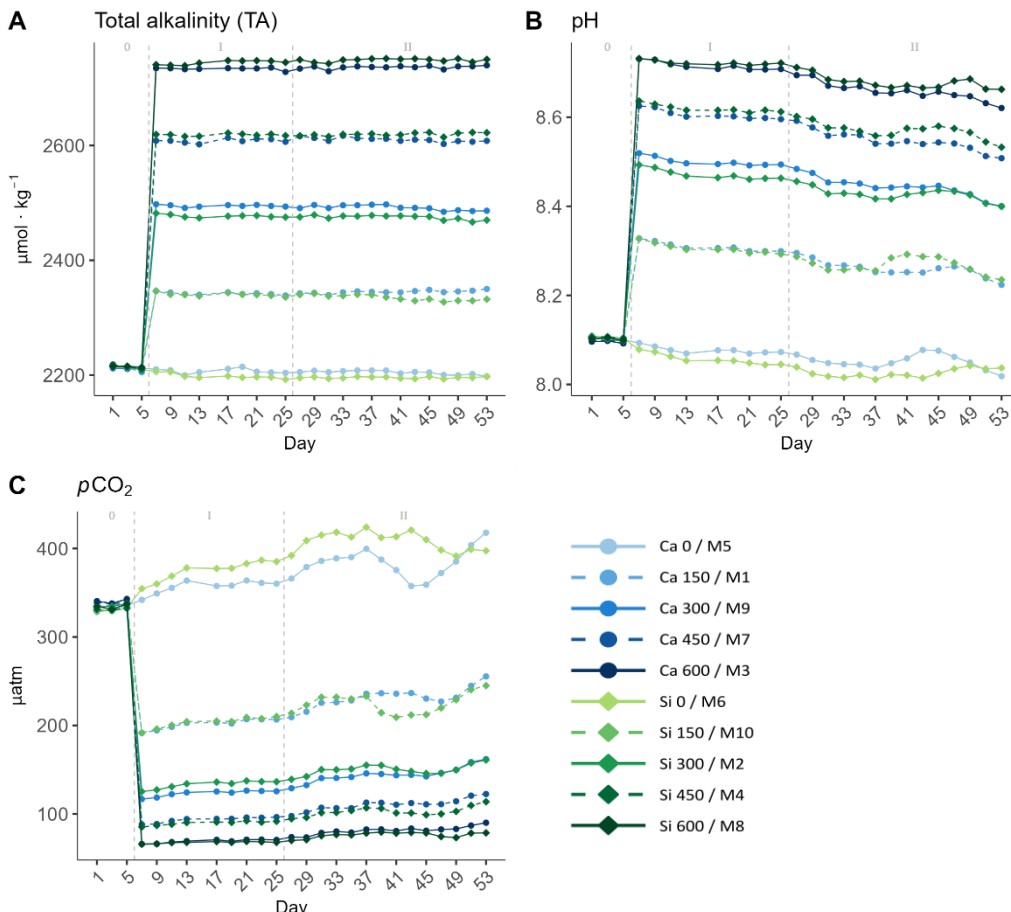

**Figure 1. Temporal development of A) total alkalinity (TA), B) pH and C) partial pressure of CO₂ ($pCO_2$) attained throughout the experiment. In the legend, the blue gradient corresponds to the calcium (Ca) treatments, and the green gradient to the silicate (Si) based ones, in both cases followed by the target delta TA levels. The grey dotted lines in all the graphs mark the (left) TA addition on day 6 and the (right) nutrient fertilization on day 26. The numbers at the top of each graph refer to the phases defined by these two additions.**

Conditions at the beginning of the experiment were consistent with those typically observed in a

280 post-bloom scenario, characterized by markedly low nutrient concentrations (Figure 2). In fact, all measured nutrients presented values below or close to the detection limit during phase I. The mean $NO_3^-$ concentration between days 7 and 25 was $0.004 \pm 0.035$ µM, $PO_4^{-3}$ ranged from 0 to 0.09 µM, and $Si(OH)_4$, in the calcium treatments, between 0.06 and 0.44 µM . In contrast, $Si(OH)_4$ concentration in the silicate based treatments, also in phase I, ranged from 65.9 to 69.8 µM. No

$Si(OH)_4$ uptake was observed during this phase. After the nutrient addition on day 26, that depicts the start of phase II, a second addition was carried out on day 28 to correct for stoichiometric differences between mesocosms (Supp. Fig. S1 and Figure 2). Mean $Si(OH)_4$ (in the calcium

treatments, no further $Si(OH)_4$ was added in the silicate treatments), $NO_3^-$, and $PO_4^{-3}$ concentrations after addition on day 28 were of $1.07 \pm 0.06$, $3.62 \pm 0.1$, and $0.2 \pm 0.02$ µM respectively.

The average concentrations of the three inorganic nutrients across the silicate and calcium treatments separately on day 29 and on the last day of the experiment, were calculated. The average concentrations for the last day were subtracted from those obtained on day 29 for the two mineral treatments separately. In the calcium treatments, $1.03 \pm 0.06$ µM, $3.56 \pm 0.09$ µM and $0.19 \pm 0.01$ µM of $Si(OH)_4$, $NO_3^-$, and $PO_4^{-3}$ respectively, were consumed during phase II. In the silicate ones, the decrease in the average $NO_3^-$, and $PO_4^{-3}$ concentrations observed across all treatments was highly similar ($3.67 \pm 0.09$ µM and $0.19 \pm 0.02$ µM). However, a steeper decrease of $10.79 \pm 1.5$ µM of $Si(OH)_4$, compared to that observed for the calcium treatments, was detected. Additionally, silicate uptake started to occur later in phase II in the calcium treatments than in the silicate ones. For further details on nutrient dynamics see Ferderer et al. (2023).

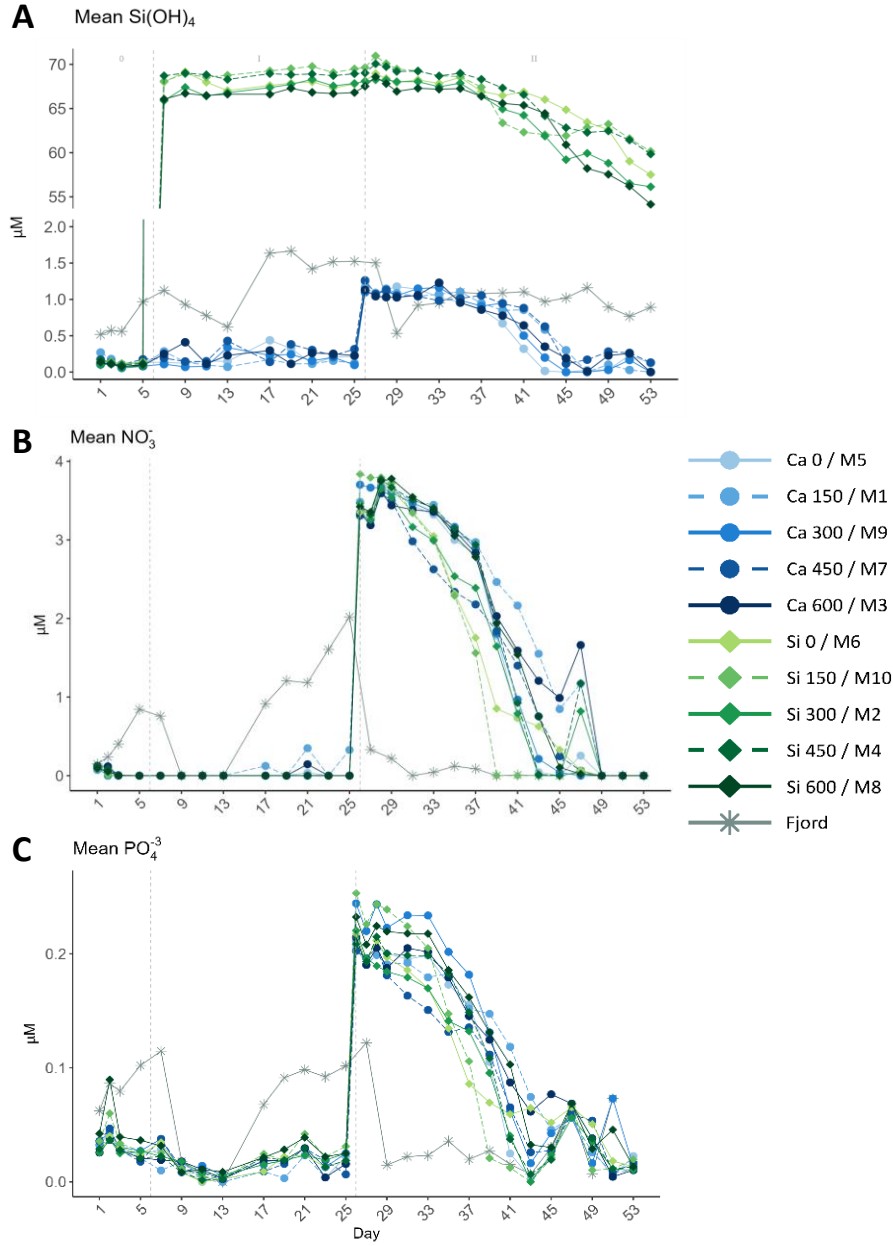

**Figure 2. Temporal development of the mean A) silicate (Si(OH)₄), B) nitrate (NO₃⁻) and C) phosphate (PO₄⁻³) obtained from triplicate daily measurements throughout the experiment. In the legend, the blue gradient corresponds to the calcium (Ca) treatments, and the green gradient to the silicate (Si) based ones, in both cases followed by the target delta total alkalinity (TA) levels. The grey dotted lines in all the graphs mark the (left) TA addition on day 6 and the (right) nutrient fertilization on day 26. The numbers at the top of each graph refer to the phases defined by these two additions.**

## 3.2 Metabolic rates and chlorophyll *a* concentration

After mesocosm enclosure, GP, and Chl*a* decreased by about half from $5.31 \pm 0.8$ μmol O₂ · L⁻¹ d⁻¹, and $1.08 \pm 0.2$ μg · L⁻¹ on day 1, to $2.43 \pm 0.4$ μmol O₂ · L⁻¹ d⁻¹, and $0.58 \pm 0.1$ μg · L⁻¹ on day 9, right after the TA addition, respectively (Figure 3). No significant TA effect was found in this short-term response phase, nor was there a difference between the two sets of mineral treatments (Figure 3 and 4). After a small rise during days 11-13, and in concordance with nutrient availability, both GP and Chl*a* continued to fall towards the end of phase I on day 25. CR showed

a similar pattern during this phase. It decreased from $1.57 \pm 0.3$ µmol $O_2 \cdot L^{-1}$ $d^{-1}$ on day 1, to $0.98 \pm 0.5$ µmol $O_2 \cdot L^{-1}$ $d^{-1}$ on day 9, yet it recovered fully by day 13. Afterwards, CR rates decreased throughout the first phase and the beginning of the second, reaching values below 1 µmol $O_2 \cdot L^{-1}$ $d^{-1}$ on day 25-27.

In the second phase, after the nutrient addition, GP rates and Chl*a* concentrations in the
mesocosms increased, reaching maximum values between days 37 and 47 (Figure 3 and 4). When looking at Figure 3, no trend can be devised. However, a discrepancy between the treatments reaching maximum values was noticed when looking at the mineral treatments (Figure 4), and at the ΔTA equivalents (Supp. Fig. S3) separately. It was observed that the low TA treatments responded to the nutrient addition slightly earlier than the high TA ones in both mineral treatment
sets (Figure 4 and S3). Low TA treatments not only responded earlier, but also reached mildly higher GP and Chl*a* values, except for the silicate control and the Δ150 µmol $\cdot$ $L^{-1}$ calcium treatment (Figure 4 and S3). As a result, we observed a progressive change in the direction of the daily linear model slopes (Figure S4 and S5). When low TA treatments began to increase until they peaked (i.e., from day 31 to 37 in the silicate, and from day 37 to 41 in the calcium treatments)
negative slopes were obtained (Figure 4, S4 and S5). Thereafter, due to the decreasing productivity in the low TA treatments and the increase in the high ones, slopes progressively became positive reaching maximum values on day 43 and 47 when the highest silicate and calcium-based TA treatments, M8 and 3 respectively, presented their maximum GP values (Figure 4, S4 and S5).

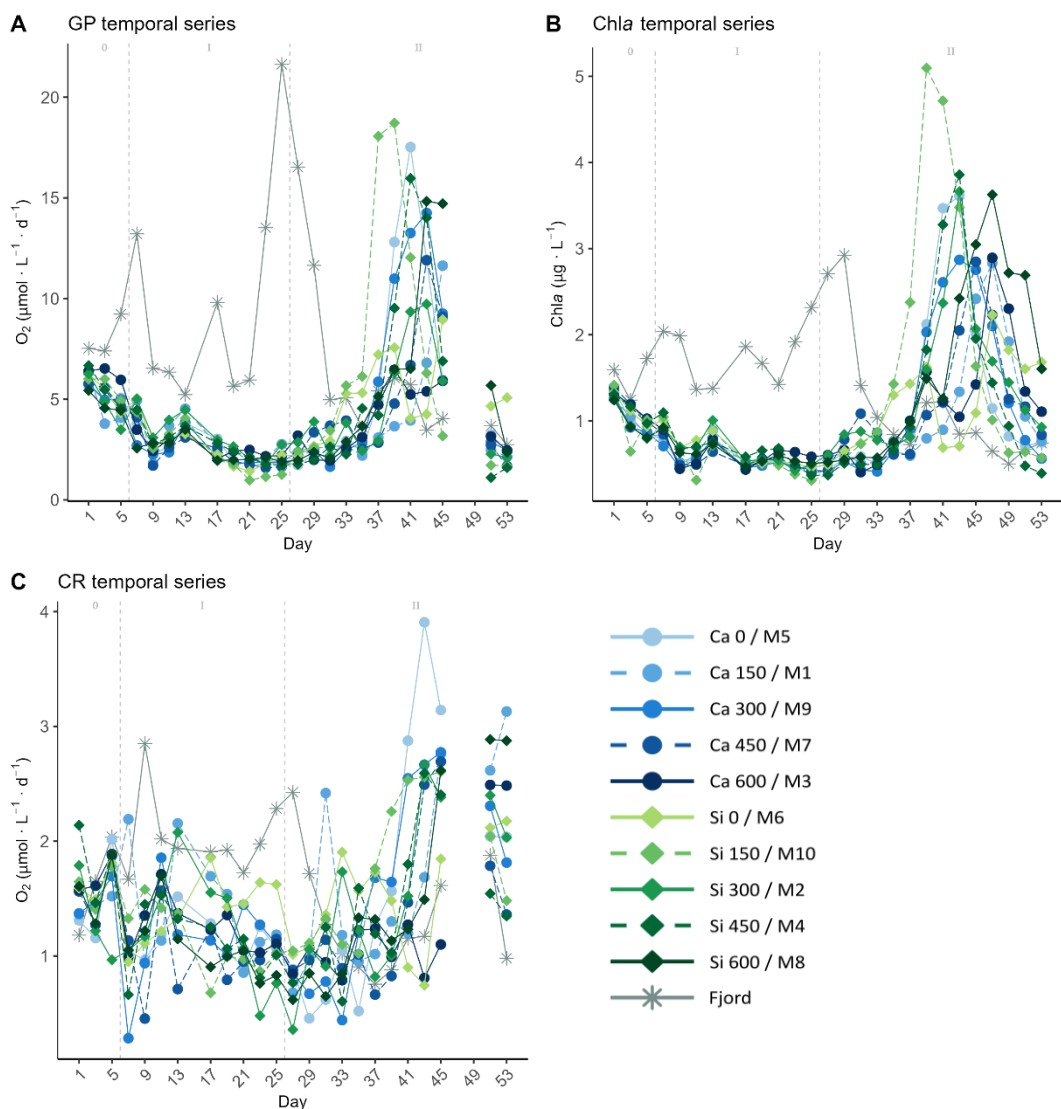

**Figure 3. Temporal developments of the measured A) gross production (GP), B) chlorophyll *a* (Chla) concentrations and C) community respiration (CR), throughout the experiment. In the legend, the blue gradient corresponds to the calcium (Ca) treatments, and the green gradient to the silicate (Si) based ones, in both cases followed by the target delta total alkalinity (TA) levels. The grey dotted lines in all the graphs mark the (left) TA addition on day 6 and the (right) nutrient fertilization on day 26. The numbers at the top of each graph refer to the phases defined by these two additions.**

When comparing mineral treatments, we observed that both GP rates and Chl*a* concentrations were marginally higher in the silicate-based than in the calcium-based treatments (Figure 4 and S3). Furthermore, as mentioned earlier, they responded slightly earlier to the nutrient addition in the silicate treatments than in the calcium ones. Indeed, the increase in GP in the silicate-based

treatment with the highest TA coincided with the peak in GP in the calcium treatments with the lowest TA (Figure 4 and S3). This was supported by the trend in, for instance, Si(OH)$_4$ concentration's temporal development (Figure 2 A), but also by the fact that while negative slopes were observed for silicate treatments from day 31 on, they were not detected for the calcium set until day 37 (Figure 4, S4, and S5).

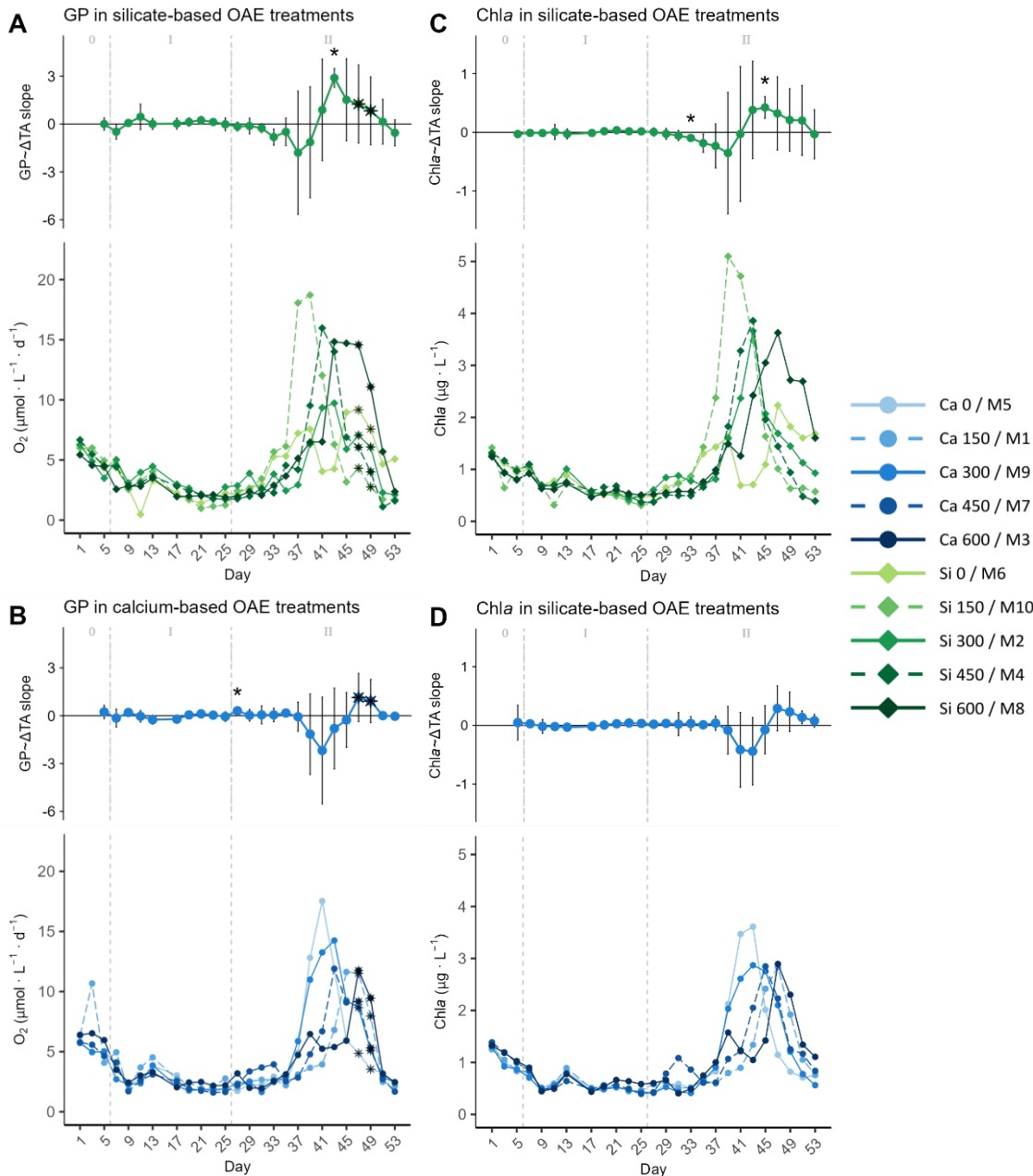

**Figure 4. Temporal development of the (top under each letter) delta total alkalinity (TA) effect size (slope ± 95% confidence intervals of daily linear models, Figure S4 and S5) on, and (bottom within each panel) absolute measurements of (A and B) gross production (GP) and (C and D) chlorophyll *a* (Chl*a*), separated by mineral treatment, where the silicate-based treatments are represented in A and C, and the calcium ones in B and D. The days when a significant relationship (p-values < 0.05) was observed are indicated with a star above the CI bars. The data points marked with a black star on top (days 47 and 49) were estimated by correlating the base-10 logarithm of GP and base-10 logarithm of Chl*a* and using the spearman model equation to calculate the missing GP values (Supp. Fig. S2). In the legend, the blue gradient corresponds to the calcium (Ca) treatments, and the green gradient to the silicate (Si) based ones, in both cases followed by the target delta TA levels. The grey dotted lines in all the graphs mark the (left) TA addition on day 6 and the (right) nutrient fertilization on day 26. The numbers at the top of the size effect graphs refer to the phases defined by these two additions.**

Although samples for oxygen-based metabolic rates were not collected on days 47 and 49 due to the COVID outbreak, it was possible to obtain data for Chl*a*. During these days, the Chl*a* concentration in the high TA calcium treatments increased, inverting the relationship with the ΔTA gradient as in the silicate-based treatments. The measured Chl*a* concentrations were correlated

with the GP rates and the model equation (Supp. Fig. S2) was used to calculate the missing GP
data points. The calculated values are marked with black stars in the figures. To calculate the
missing CR values, the measured GP and NCP rates were correlated, so that the missing NCP
rates could be calculated with the latter model equation (Supp. Fig. S2), and then subtracted from
the previously estimated GP. CR contributed significantly less than NCP to GP during the second
phase (Figure 3 to 5).

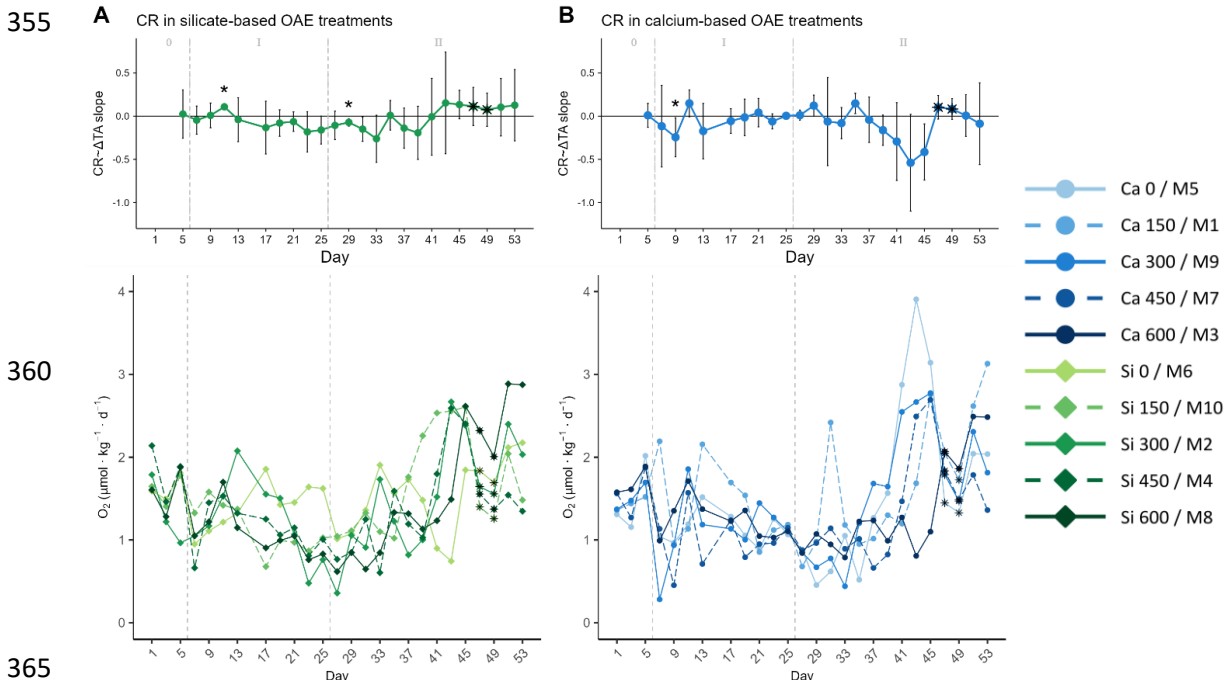

**Figure 5. Temporal development of the delta total alkalinity (TA) effect size (slope ± 95% confidence intervals of daily linear models, Figure S6) on, and (bottom within each panel) the temporal development of the absolute community respiration (CR) rates for the (A) silicate and (B) calcium-based treatments. The days when a significant relationship (p-values < 0.05) was observed are indicated with a star above the CI bars. The data points marked with a black star on top (days 47 and 49) were estimated by correlating the measured net community production (NCP) to the gross production (GP), then using the obtained spearmen model equation to calculate the missing NCP values, and finally subtracting the latter to the calculated GP values. These were estimated using the spearman correlation equation of base-10 logarithm of the measured GP and base-10 logarithm of Chla. The vertical grey dotted lines in all the graphs mark the (left) TA addition on day 6 and the (right) nutrient fertilization on day 26. The numbers at the top of each graph refer to the phases defined by these two additions.**

CR rates in the second phase did not portray the aforementioned pattern followed by the GP rates
and the Chl*a* concentrations (Figure 5A). They showed an increase after day 41, probably fueled
by the organic matter produced during phytoplankton growth. Indeed, the progression from
negative to positive slopes observed also for CR, as well as the earlier response of silicate versus
calcium-based treatments support the idea that high respiration rates were sustained by primary
production (Figure 5).

Outside the mesocosms, in the fjord, conditions were different. GP rates and Chl*a* concentration
remained close to the initial values, showing three peaks around days 7, 17, and 25, fueled by
nutrient inputs (Figure 2). Of the three peaks, the last one was the most intense reaching over 20
μmol $O_2 \cdot L^{-1} d^{-1}$, and up to 3 μg $\cdot L^{-1}$ of Chl*a*, respectively. This bloom supposed the depletion of

NO$_3^-$ and PO$_4^{-3}$ (Figure 2), and thus the collapse of the bloom by day 33. Both GP and Chl*a* continued decreasing to values below initial conditions towards the end of the experiment. A similar pattern was showed by CR in the fjord, with the last peak reaching almost 2.5 µmol O$_2$ ·

L$^{-1}$ d$^{-1}$, yet no increase was observed on day 17. Furthermore, CR values were of the same magnitude as the initial ones from the end of the bloom towards the end of the experiment (Figure 3 C).

### 3.3 Metabolic balance and assimilation numbers

The absolute metabolic balance (GP:CR ratio) in both treatments presented a similar trend

throughout the experiment (Figure 6). Values were consistently above 1 indicating that respiration was greater than production, i.e., the community was in an autotrophic state, even during phase I when low nutrient availability inside of the mesocosm constrained primary productivity. Due to nutrient starvation in phase I and the consequent decrease of GP, GP:CR presented its minimum on day 21-25. After the nutrient addition, GP:CR increased, reaching maximum values between

400    days 33 and 45. Afterwards, GP declined but CR was enhanced leading to a new decrease in GP:CR. It should be noted that in the last 2 days of the experiment, GP:CR ratio presented values below 1, i.e., the systems became heterotrophic after phase II blooms.

Overall, comparing both treatments, slightly higher values were observed in the silicate treatments than in the calcium ones, being more pronounced during bloom development in phase II. The

between peak delay pattern observed in GP and Chl*a* was only apparent for the silicate treatments (Figure 6 A).

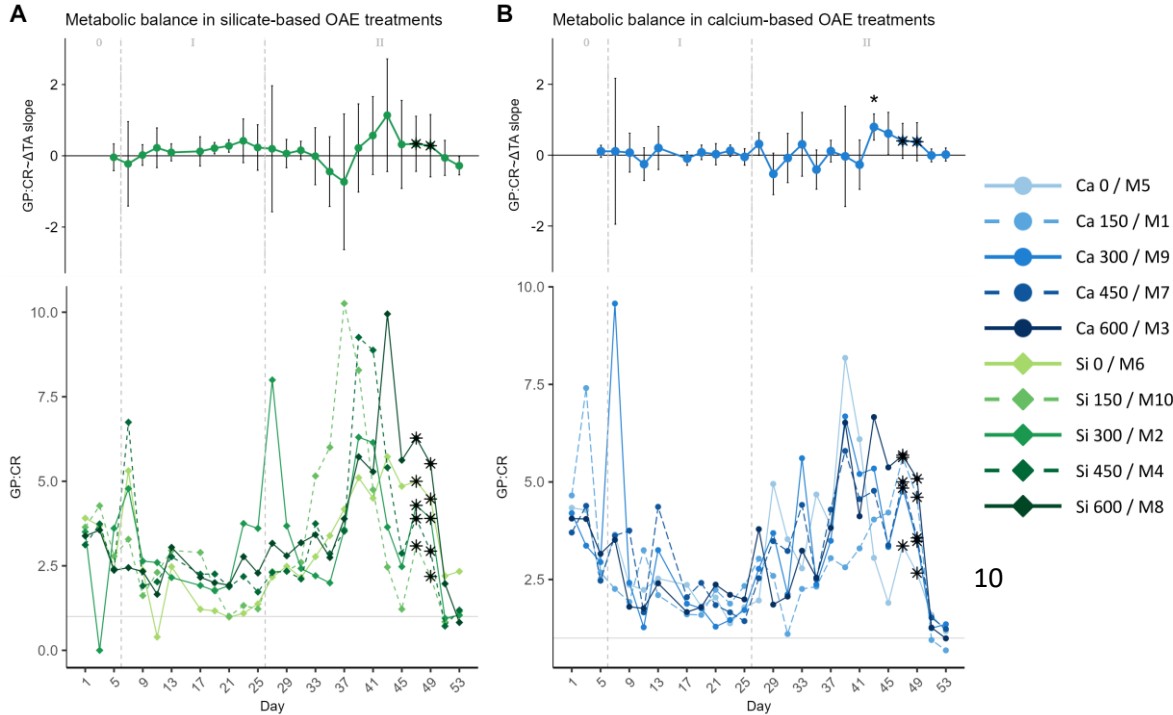

**Figure 6. Temporal development of the (top under each letter) Δtotal alkalinity (TA) effect size (slope ± 95% confidence intervals of daily linear models, Figure S7) on, and (bottom within each panel) of the calculated metabolic balance (ratio of gross production, GP, over community respiration, CR), where A) represents the silicate-based treatments, and B) the calcium ones. The days when a significant relationship (p-values < 0.05) was observed are indicated with a star above the CI bars. The data points marked with a black star on top (days 47 and 49) were calculated by dividing estimations of GP and CR. The missing GP values were estimated using the spearman correlation equation of the base-10 logarithm of GP and base-10 logarithm of Chl*a*. Then CR were calculated by correlating the measured Net Community Production (NCP) to the Gross Production (GP), then using the obtained spearmen model equation to calculate the missing NCP values, and finally subtracting the latter to the calculated GP values. In the legend, the blue gradient corresponds to the Calcium (Ca) treatments, and the green gradient to the Silicate (Si) based ones, in both cases followed by the target delta TA levels. The grey dotted lines in all the graphs mark the (left) TA addition on day 6 and the (right) nutrient fertilization on day 26. The numbers at the top of the size effect graphs refer to the phases defined by these two additions.**

Assimilation numbers remained reasonably constant throughout the experiment and unaffected either by the mineral treatment, or by the TA gradient (Figure 7). Differences between phases were not apparent either. In the fjord though, the assimilation numbers in the last bloom on day 415 25 reached a maximum that was higher than those observed in the mesocosms throughout phase II (see temporal series in Figure 7).

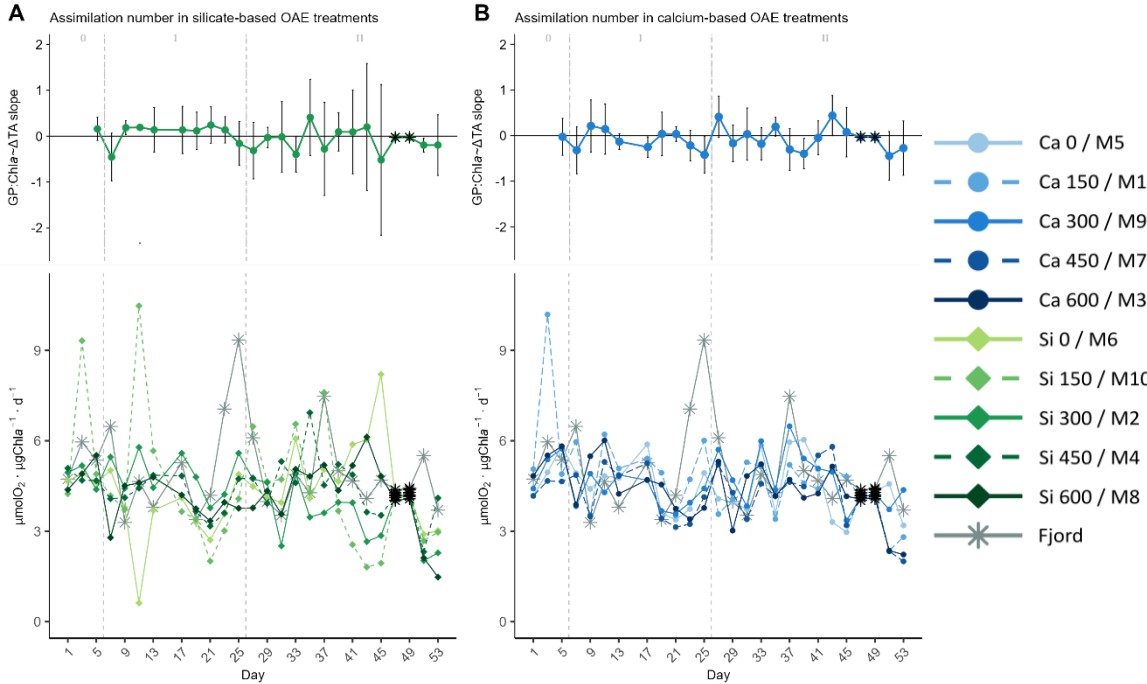

**Figure 7. Temporal development of the (top under each letter) delta total alkalinity (TA) effect size (slope ± 95% confidence intervals of daily linear models. Figure S8) on, and (bottom within each panel) of the calculated assimilation numbers (ratio of gross production, GP, over chlorophyll *a*, Chl*a*, concentration), where A) represents the silicate-based treatments, and B) the calcium ones. The data points marked with a black star on top (days 47 and 49) were calculated because GP could not be measured on those days. The missing GP values were estimated using the spearman correlation equation of the base-10 logarithm of GP and base-10 logarithm of Chl*a*. In the legend, the blue gradient corresponds to the calcium (Ca) treatments, and the green gradient to the silicate (Si) based ones, in both cases followed by the target delta TA levels. The grey dotted lines in all the graphs mark the (left) TA addition on day 6 and the (right) nutrient fertilization on day 26. The numbers at the top of each graph refer to the phases defined by these two additions.**

In all the previous Figures portraying the effect size of ΔTA over time, statistical significance (p value < 0.05) was indicated with a star on top (*) of the days when it was observed. The 95% confidence intervals are wide, suggesting high variability on individual days during phase II. The two treatments that deviate from the trend described in terms of GP and Chl*a* are the control in the silicate treatment set, and the Δ150 in the calcium one. The former portrayed two small peaks during the bloom phase and the highest CR rates right before the addition (Figure 5 A, bottom), and the latter the lowest GP:CR values in the second phase (Figure 6 B, bottom), and a longer delay than the control (Supp Fig. S4 and S5 respectively). Due to the small sample size of the mineral treatments (n = 5) on each individual day, these figures are used to portray the observed pattern more clearly. Not to summarize statistical outputs.

## 4. DISCUSSION

The experiment described herein involved the application of two distinct alkalinity gradients utilizing two ocean alkalinity enhancement (OAE) strategies based on different minerals: forsterite ($Mg_2SiO_4$) and hydrated lime ($Ca(OH)_2$). Forsterite is the Mg endmember of olivine,

which occurs commonly in nature, while hydrated lime is produced through the calcination of limestone (Renforth and Henderson, 2017). Both minerals have been considered for OAE implementation; however, experimental studies on their potential impacts on natural communities remain unaddressed. In the current study, simulations were carried out by, instead of using the raw minerals, diluting the elements that conform them separately, alongside NaOH, to reduce the potential effects of confounding variables such as the release of raw mineral impurities.

Large scale equilibrated OAE deployment would require the use of reactors for pre-equilibration (Hartmann et al., 2023) so it would be much more costly and less scalable. Therefore, $CO_2$ sequestration was not conducted prior to the alkalinity manipulation to evaluate the potential effects of the carbonate chemistry alterations associated with a non-equilibrated OAE deployment. These entailed a persistent increase in pH and decrease in $p$CO$_2$ when compared to ambient levels, since full natural equilibration throughout the experiment did not occur.

In this study, the focus was to ascertain whether the microbial community, inhabiting a neritic system, would respond in terms of production and respiration rates during a 49-day exposure to the aforementioned conditions. Besides, differences in the response to the calcium and silicate inputs caused by the mineral addition simulations were expected too. Especially in terms of community composition with calcium-based OAE treatments through the hydrated lime addition simulation (with compounds containing $Ca^{2+}$ and $OH^-$ separately), potentially increasing the abundance of pelagic calcifiers, and silicate-based OAE through the forsterite one (with compounds containing $Mg^{2+}$, $SiO_3^{2-}$, and $OH^-$ independently), favoring diatom proliferation (Bach et al., 2019b). The latter community shifts were predicted to yield distinct responses in terms of absolute community metabolic rates (gross production, GP; net community production, NCP; community respiration, CR), and production efficiency via clear differences in assimilation numbers (GP:Chl$a$) over time.

### 4.1 Response to the carbonate chemistry conditions: Non-equilibrated TA gradient

Our results showed that OAE effects on microbial metabolic rates may have been linked to nutrient availability. Under the initial low inorganic nutrient concentrations (Figure 2), characteristic of post-bloom conditions, the response of the microbial community production and respiration rates to the TA manipulation was likely concealed by the nutrient limitation. After the simulated mixing event on days 26 and 28 and the consequent nutrient enhancement however, a discernible response pattern emerged. Blooms occurred in all mesocosms despite the carbonate chemistry conditions suggesting that the phytoplankton community were resilient to a TA manipulation of up to 600 µmol L$^{-1}$. Nevertheless, the most noteworthy result of our experiment is the qualitatively inferred, mild delay in bloom formation with increasing TA.

In the past, before ocean acidification emerged as a central focus of scientific research, several culture experiments simulating high pH/low $CO_2$ conditions were carried out. For instance, Goldman (1999) carried out 12-day pH-drift, batch culture experiments, with three large diatom species (*Stephanopyxis palmeriana, Ditylum brightwelli* and *Cosinodiscus* sp.) and found that, when pH rose to above 8.5, growth rates started to decline. Similarly, Hansen (2002) performed 7-day experiments with three dinoflagellate species to evaluate their response in terms of growth rates and community succession. These were exposed to a pH range of 7.5 – 10, applied through the addition of NaOH, to simulate the conditions in the Mariager Fjord, Denmark. In these experiments, the three species' growth rates were the highest at pH 7.5, and a decrease to a 20% of the maximum growth rate was observed at pH 8.3-8.5 for *Certaium lineatum*, and at 8.8–8.9 for *Heterocapsa triqueta* and *Prorocentrum minimum.* The latter also showed different tolerance limits, specifically 8.8, 9.5, and 9.6, respectively.

Additionally, despite the lack of data from experiments performed at constant pH at the time, Hansen (2002) compared his results with literature data on 35 species of marine phytoplankton and concluded that there was high variability in their tolerance to high pH (8.4-10). Even within the same family, species-specific tolerance limits could be observed. Most of these species would have been able to grow past the levels reached in the present study (half could grow above pH 9.2), but their growth rates would decrease, at different species-specific levels, in relation to the increased pH (Chen & Durbin, 1994; Hansen, 2002). Therefore, offering the main explanation behind the qualitatively described delay in bloom formation. Although information on the species' composition will be essential to fully evaluate the latter trend.

High extracellular pH can alter key membrane transport processes and metabolic functions involved in pH regulation (Smith and Raven, 1979). It can also induce changes in cellular amino acid relative composition and overall content (Søderberg and Hansen, 2007). But a key variable behind the response in terms of production observed in the current study is the inter-speciation of inorganic carbon associated with the non-equilibrated nature of the TA manipulation. At pH 8, ~ 1% of the DIC in seawater is present as $CO_2$, but at pH 9, this percentage is reduced 10 times over (Hansen, 2002). Additionally, 95% of the carbon fixation in the ocean is undertaken by the photosynthetic carbon-fixation reaction mediated by the RUBISCO (ribulose bisphosphate carboxylase oxygenase) enzyme, which can only employ $CO_2$ (Raven, 2000). Thus, besides the pH levels reached here being around 8.7 in the highest treatments (calcium and silicate $\Delta 600$ µmol $\cdot$ L$^{-1}$), which were well above the maximum of 8.2 measured by Omar et al. (2019) in early spring (from 2005 to 2009) along the Norwegian west coast (specifically in Raunefjorden, our study site's location, as well as in Korsfjorden, Langenuen, and southern parts of the Hardangerfjorden), the associated low $p$CO$_2$ attained was likely the detrimental treatment variable. Notably, the Chl$a$ concentration, a proxy for phytoplankton biomass, was higher inside the mesocosms than in the

fjord during phase II (Figure 3). This suggests that certain species may have thrived under the altered carbonate chemistry conditions present in the mesocosms. However, the microbial community in these mesocosms exhibited lower oxygen production per unit of Chl*a*, indicating that, despite an increase in phytoplankton biomass, the microbial community under high pH/low $p$CO$_2$ conditions may have been less efficient in terms of production than the one in the fjord (Figure 7).

Although CO$_2$ is the main substrate for photosynthesis, in its absence, many marine phytoplankton species adapt by using HCO$_3^-$ through carbon concentrating mechanisms (CCMs; Giordano et al., 2005). The latter are more energetically costly because they entail the concentration of HCO$_3^-$ at the diffusion range of the plasma membrane, and the catalysis of HCO$_3^-$ to CO$_2$ for posterior CO$_2$ fixation. Thus, these pathways are less efficient, causing reduced growth rates, and the subsequent delay in bloom formation with increasing TA without pre-equilibration (higher pH/lower CO$_2$ conditions). Although the response pattern found was described qualitatively due to the lack of statistical strength (small sample size of the mineral treatments, n = 5, when considered separately) in the experimental design, it could be a relevant finding, granting further evaluation.

**4.2 Response to the two simulated mineral additions: Calcium- vs silicate-based**

The observed delay was, although not significantly, longer in the calcium than in the silicate treatments. Meaning that the low silicate treatments responded slightly earlier to the nutrient addition, than the low calcium ones. Besides the three-day lag noticed for the calcium treatments' response to the nutrient addition with respect to that of the silicate ones, and despite assimilation numbers showing no response to the TA gradient in either case, several differences between the two mineral treatments were noticed. Mildly higher production rates and Chl*a* concentration were observed in the silicate treatments when compared to the calcium ones. Both silicate and calcium treatment sets' microphytoplankton fractions included diatoms, in different proportions based on the mineral treatment, during the second phase (Ferderer et al., 2023; Kittu et al., 2024, in prep).

Small diatoms are usually well adapted to post-bloom conditions due to their favorable surface to volume ratio, an adaptation to low nutrient concentrations (Raven, 1986), and their well-developed CCMs (Chrachri et al., 2018). The silicate uptake in the silicate treatments during phase II was of Si:N ~10.7: 3.7 µM, suggesting the community may have been diatom dominated. While in the calcium treatments, silicate to nitrate were consumed in a ~1: 3.6 µM ratio. Diatoms have been observed to reduce their silicon requirements by exhibiting frustule thinning if exposed to silicate-limiting conditions to prioritize growth (McNair et al., 2018). In fact, Ferderer et al. (2023) found that diatoms in the silicate treatments were more heavily silicified than in the calcium ones, generally independently of the TA level. Diatoms in these treatments may thus have been able to allocate more resources to growth and photosynthesis (Inomura et al., 2023).

Therefore, the excess availability of silicate, coupled with acclimation to the carbonate chemistry during phase I, could have aided the faster response to the mixing event simulation, as well as contributed to the higher absolute GP and Chl*a* concentrations observed in the silicate treatments.

Further differences between mineral treatments could be inferred in the effect size of ΔTA on CR rates over time, which translated to differences in the effect of ΔTA on metabolic balance (GP:CR), in the second phase. A potential explanation for these results is likely in relation to differences in the phytoplankton community composition between the two sets of treatments. Notably, Ferderer et al. (2023) found that the effect of the mineral on silicification varied among diatom genera in the same experiment. Furthermore, even if the community included diatoms, rapid diatom growth during a spring bloom does not necessarily suppress other non-diatom phytoplankton growth (Barber and Hiscock, 2006; Lochte et al., 1993). The latter may increase in absolute cell concentrations, but their absolute contribution to the total biomass when diatoms are abundantly present could be modest (Barber and Hiscock, 2006). This would therefore explain why no differences between mineral treatments can be inferred, even if they occurred, in the absolute assimilation number (GP:Chl*a*) temporal development, nor in their overtime response to the ΔTA gradient (Figure 7). It is though likely that the non-diatom phytoplankton community, both in terms of composition and contribution, differed between mineral treatments.

Lastly, the silicate control did not behave as expected. The increase in production and in Chl*a* concentrations in the form of a large peak, as in all other treatments after the inorganic nutrient addition, was not observed. Instead, both in terms of GP and Chl*a*, two small peaks, a half in magnitude when compared to the calcium control, were noted. The observed discrepancy between the controls was unanticipated and may have resulted from a random response attributable to the mesocosm effect. Indeed, in this treatment, the absolute CR (Figure 5) and its contribution to GP right before the nutrient addition was the highest when compared to the rest. When communities are enclosed, even if initially they are very similar, they are not the same and, over time, they may behave differently. The pre-addition phase allows for acclimation, though it is held as short as possible to keep a similar community when the treatment is applied. However, it is still a bias to be taken into consideration.

### 4.3 Potential implications and future research

Given the novelty of this field and the findings from the current study, we outline some of the remaining unresolved research questions. Trophic decoupling between phytoplankton and zooplankton has been observed in a large temperate lake where *Daphnia* resting eggs were unaffected by temperature increase while the phytoplankton spring bloom occurred earlier in the year (Winder and Schindler, 2004). This led to a long-term decline of *Daphnia,* the keystone herbivorous zooplankton species, due to the increased mismatch with the spring bloom. The

opposite could be expected due to the observed delay in bloom development associated with the non-equilibrated OAE implementation. That is if micro- and mesozooplankton do not respond to the pH levels attained themselves, favoring trophic transfer. Considering that the transport of particulate organic matter to depth is regulated by the coupling of primary and secondary producers (Wassmann, 1998), thus potentially affecting the OAE's CDR efficiency, this is a key implication that may require further assessment.

Additionally, if the carbonate chemistry conditions were altered under pre-bloom or blooming conditions, the initial shock without prior acclimation may be stronger. It could potentially translate to longer delays in bloom development with increasing TA, and even changes in community composition and overall production efficiency at certain alkalinity levels. Consequently, in turn possibly affecting trophic transfer and carbon export.

Finally, we hypothesize that the delayed response observed in relation to the TA manipulation would likely be amplified if higher non-equilibrated alkalinity additions were deployed. Although, in real-world applications, carbonate chemistry alterations may be transient and less extreme, experiments that look into i) how and when absolute community metabolic rates respond to higher TA levels, ii) where the threshold in the higher range of the gradient at which reduced GP:Chl$a$ ratios could be induced is, but also iii) if a recovery could occur after a long-term exposure, or after dilution (short-term), are essential if safe deployment limits are to be identified.

## 5. CONCLUSIONS

This study presents the first experimental evaluation of the effects of a non-equilibrated, silicate vs calcium-based OAE deployment under natural conditions and at a mesocosm scale on microbial metabolic rates. The total alkalinity (TA) manipulation (a silicate and a calcium based five-step $\Delta$TA gradients 0 – 600 $\mu$mol $\cdot$ L$^{-1}$), without prior $CO_2$ sequestration, resulted in a stable increase in pH and a decrease in $pCO_2$, that persisted until the end of the experiment. Conducted in a neritic system under post-bloom conditions, a mixing event was simulated roughly halfway through it. Following the inorganic nutrient addition, a mild delay in bloom formation, based on gross and net community production (GP and NCP) rates, and chlorophyll $a$ (Chl$a$) concentrations, in relation to the $\Delta$TA gradient, was observed. Low TA treatments responded slightly earlier than high TA ones, with the delay being longer for the calcium compared to the silicate treatments. This delay is likely linked to the previously reported, species-specific negative relationship between high pH/low $pCO_2$ conditions and phytoplankton growth rates. Although, overall, phytoplankton were resilient to the TA levels attained.

Seasonal differences in this pattern require further investigation, as nutrient limitation may have concealed a potentially more pronounced short-term response to the TA addition. Given the

qualitative nature of the described trend, additional experiments testing it further through, for instance, higher resolution gradients, are necessary. Further, while real-world OAE applications

are expected to result in more transient and less extreme carbonate chemistry perturbations, a thorough assessment of TA limits is crucial. Particularly because the detected response may become more pronounced at higher TA levels and vary depending on microbial species composition. Therefore, studies evaluating recovery from intense carbonate chemistry perturbations will be essential to establish safe TA addition limits for future OAE implementation.

**Data availability**

The dataset of the metabolic rates used in this study can be found in the online repository PANGAEA. Data can be accessed via the following link: https://doi.org/10.1594/PANGAEA.972371 (Marín-Samper et al., 2024b). Chlorophyll *a*,

carbonate chemistry and inorganic nutrient concentration data will be made available in the same repository.

**Author contributions**

Experimental concept and design: UR and JA. Direct participation in the experiment: LMS, JA

and UR. Data analyses: LMS with input from NHH. Original draft preparation: LMS. Review and editing: All authors.

**Financial support**

This research has been supported by the Horizon 2020 Research and Innovation Programme

project OceanNETs ("Ocean-based Negative Emissions Technologies – analysing the feasibility, risks and cobenefits of ocean-based negative emission technologies for stabilizing the climate", grant no. 869357), and by the EU H2020-INFRAIA's project AQUACOSM ("AQUACOSM: Network of Leading European AQUAtic MesoCOSM Facilities Connecting Mountains to Oceans from the Arctic to the Mediterranean", Project No.: 731065). Further, it was co-financed by the

"Agencia Canaria de Investigación, Innovación y Sociedad de la Información" (ACIISI) of the "Consejería de Economía, Conocimiento y Empleo", and by the "Fondo Social Europeo (FSE) Programa Operativo Integrado de Canarias 2014-2020, Eje 3 Tema Prioritario 74 (85%)".

**Competing interests**

The authors declare that they have no conflict of interest.

**Acknowledgements**

The authors would like to express their most sincere gratitude to the entire KOSMOS team at GEOMAR for their invaluable support in managing logistics and, to the technical team for all the hard work building and maintaining the mesocosms during the campaign. Special appreciation is extended to their dedication in coordinating on-site research activities and fostering fair data management and exchange. A special thank you goes to Acorayda González from the biological oceanography group (GOB-ULPGC), for helping with the oxygen measurements. Also, we would like to commend Julieta Schneider (GEOMAR) and Dr. Kai Shulz (Southern Cross University) for the carbonate chemistry measurements, Juliane Tammen for the inorganic nutrient concentration data, Dr. Leila Kittu and Levka Hansen for the chlorophyll a data, and Dr. Leila Kittu (GEOMAR) Xiaoke Xin, Phillip Süßle, and Dr. Joaquin Ortiz for the interesting discussions on data interpretation. We also want to thank the University of Bergen for providing the Espegrend Marine Research Field Station to conduct the experiment, and the station's staff for all their assistance throughout. Lastly, we would like to acknowledge the three referees and the associate editor, Dr. Jack Middleburg, that contributed to improving this manuscript's quality through their insightful feedback.

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
