# Peer review of "Responses of Microbial Metabolic Rates to Non-Equilibrated Silicate versus Calcium-Based Ocean Alkalinity Enhancement"

_EGUsphere, 2024_

## Author Comment (AC2)

**RC1 Specific comments addressed:**

The authors presented a comprehensive mesocosm study in a mid-latitude fjord that employed two types of ocean alkalinity enhancement (OAE) techniques to study the changes in carbonate chemistry on metabolic rates in the experimental mesocosms. Different levels of OAE were tested and delays in phytoplankton bloom compared to control conditions were revealed. This study did not use the common OAE minerals (hydrated lime and olivine) directly but employed chemical additions that mimicked the outcome of applying these minerals hence other confounding factors such as trace metal release can be avoided, which is a clever design.

The manuscript is mostly well written, but it can be verbose in places. First, here is a technical question that I hope the authors can address. In this study nutrient sample collection and processing, the authors used 0.45 μm filters for nutrient sample collection and then the samples were kept in the dark at ambient temperature until further processing (did you mean analysis)? See Line 165-166. Given the fact that nutrient stoichiometry is important in discussing metabolism in the Ca vs. Si based OAE schemes, this nutrient collection technique needs further clarification, and the authors should affirm that the pore size and sample preservation had not inadvertently altered nutrient concentrations. See below for a reference.

Reed, M.H., Strope, E.K., Cremona, F., Myers, J.A., Newell, S.E. and McCarthy, M.J., 2023. Effects of filtration timing and pore size on measured nutrient concentrations in environmental water samples. Limnology and Oceanography: Methods, 21, 1-12.

*The article provided mentions that, in terms of pore size, all nutrient samples should be filtered through maximum 0.45 μm. Our samples were collected in triplicate and filtered through 0.45 μm 1-2 hours after collection (added to the manuscript) and stored in the dark until they were analysed. They were actually stored in the fridge after filtration, which has been included in the manuscript. A piece of information missing relevant to this method's reliability is the time between filtration and analyses, which was of less than 6 hours. This information has also been added to the manuscript. The method employed to measure Si(OH)4, NO3, NO2, and PO4 concentrations, and the one followed to measure NH4, come from two publications, specifically from* Hansen & Koroleff (1999) *and* Holmes et al. (1999)*, which have been cited over 1200 and 1400 times, respectively.*

*Additionally, Reed, et al. (2023) evaluate how higher pore sizes and longer times until analysis reduce the PO4 and NH4 determination reliability, particularly when baseline nutrient concentrations are very low. Our discussion section on nutrient uptake focuses on Si to N (derived from NO3 uptake, not NH4) ratios, after a significant nutrient addition. Besides, after the nutrient addition, the measured concentrations were consistent with the theoretically intended ones.*

Below are some minor comments:

1. Be consistent with the descriptions of the duration of the experiment. 10-weeks (Line 92), 53 days (Line 127), and three-month (Line 369) were all used.

   *Addressed. They were all changed to 53-day.*

2. There are many places where the words "said", "mentioned", "aforementioned", "particular", "present" etc were used and in most cases these words are either unnecessary or confusing. Please remove or reword.

   *Addressed. These terms were mostly removed across the entire manuscript. And were present is used as in "the present study" (referring to ours), we now use "current".*

3. Throughout the context, while it is understandable that a calcium-based chemical alternation was made to the experimental system, using "calcium" appears a little misleading because both OAE approaches intend to increase concentrations of carbonate species in the water. Silicate weathering leads to an increase in carbonate ion concentration, and hydrated lime is essentially a direct base addition, not adding calcium per se. I would suggest that the authors to reconsider the term usage.

*This terminology was mainly used to differentiate between the calcium-based and silicate-based sets of treatments, as these elements were added to test the 'green vs. white ocean' hypotheses proposed by Bach et al. (2019). In this experiment, TA was adjusted using NaOH, thus bases were added directly in both scenarios. Therefore, the main relevant difference between the two sets of treatments is the addition of calcium and silicate. This is the key difference because even though, in the silicate-based treatments, Mg was also added to simulate a forsterite addition, this element is already found in high concentrations in seawater due to its long residence time (Foster et al., 2010)*

*Furthermore, this terminology will be consistent across many publications about different parameters measured during the same mesocosm campaign that are currently in preparation. Therefore, it will aid in the intercomparison of all these publications to get the whole story of what happened in this very large and collaborative experiment.*

4. Line 44-45, improper punctuation.

*Addressed.*

5. Line 90, for an uncommon chemical/mineral, explain forsterite.

*This was addressed by adding the following information: However, olivine is comprised of forsterite ($Mg_2SiO_4$) and fayalite ($Fe_2SiO_4$) in a 9:1 ratio. An iron (Fe) addition may have a fertilizing effect on phytoplankton in the photic zone (Bach et al., 2019; Hauck et al., 2016; Renforth & Henderson, 2017), and it is the Mg end member of olivine that, as it weathers, naturally consumes atmospheric $CO_2$ (Köhler et al., 2013; Renforth & Henderson, 2017).*

6. Line 115, what's in this brine solution.

*Addressed by specifying it was a NaCl brine solution*

7. Line 162, provide more details on how pH was corrected and how the comparison looked like.

*An article focusing on the carbonate chemistry from this experiment is in preparation. Nonetheless, a reference to an article on seawater carbonate system considerations in the context of OAE research, which explains this process in more detail, has been added.*

8. Line 178 vs. Line 190, clarify whether the "initials" were already fixed before the incubation.

*Addressed.*

9. Line 180, what's "blackout"? Please use proper term/description.

*Blackout was changed to opaque.*

10. Section 2.5, more details on Chl-a processing and analysis is needed. What's the purpose of using the 200 μm mesh?

*We want to extend our gratitude to the referee for this comment because an error was detected. Samples were not pre-filtered using a 200 μm mesh (this referred to the sample processing of another parameter that is not included in the current study). We specified that samples were filtered through GFF with a 0.7 μm pore size and stored at −80 ºC until they were analysed fluorometrically the following day.*

11. Line 239-244, the sentences read awkward and confusing. Please restructure and clarify, explain what's the "controls" mean in the context of the experimental design.

*Addressed. This sentence was changed to: "Therefore, the pH and $pCO_2$ in the mesocosms where TA was manipulated did not reach ambient levels throughout the experiment."*

12. Line 251-252, remove "significantly", and did the experimental timing coincide with post bloom period in this fjord? If so, this needs to be mentioned in the method section.

*To address this comment, significantly was removed, and the requested information was added in the first sentence of the methods section: "The experiment (KOSMOS Bergen 2022) was carried out in Raunefjorden, 1.5 km offshore from the Espegrend Marine Research Field Station, of the University of Bergen, Norway, under post-bloom conditions, starting on the 7th of May 2022."*

13. Line 263, subtracted "from"?

*Addressed.*

14. Line 284, "slightly almost", what does it mean?

*We want to thank you for noticing. It was a typo. Slightly was removed*

15. Line 286-287, "little under" as "slightly below"?

*Corrected.*

16. Line 304-307, this sentence needs to be reworded as the current form is quite confusing.

*The sentence was re-written, and we hope it is clearer this way: Therefore, the increase in GP in the silicate based, highest treatments coincided with when GP peaked in the low TA calcium ones. Hence, the delay in the community's response to the nutrient addition was longer for the calcium than the silicate treatments, in both cases following the TA gradient (Figure 3A and B).*

17. Line 326-327, the sentence "Nonetheless …" is not clear.

*Addressed by specifying what pattern we were referring to: "Nonetheless, negative slopes obtained from daily linear models peaking on day 41 and that reversed on day 47, as observed in terms of GP and Chla in the calcium treatments, can be partially inferred (Figure 5B)." The following sentence was also altered to: "CR in the low TA treatments increased around the same time as in terms of GP and, the calculated CR rates that followed showed a slight recovery in the high TA treatments."*

18. Line 331, define "metabolic balance"

*After metabolic balance "GP:CR" was included in parenthesis.*

19. Line 343, "latter parameters" meaning?

*"Latter parameter" was changed to GP:CR*

20. Line 347-353, this paragraph appears fragmented and difficult to follow. Please revise.

*To clarify the content of this paragraph, it was re-written and divided in two separate paragraphs: "Furthermore, to see if the observed pattern also translated to some extent to the community composition, assimilation numbers based on the GP rates were calculated. GP was chosen due to the low and relatively constant contribution of CR, especially during the second phase. Additionally, because the NCP was positive throughout the experiment, the actual production must have been at least as much as the CR.*

*The GP normalization using Chla as a biomass proxy (GP:Chla) yielded assimilation numbers that remained reasonably constant throughout the experiment and overall unaffected either by the mineral treatment or by the TA gradient. Differences between phases were not apparent either."*

21. Line 366-367, "persistent increase in pH and decrease in pCO2" needs proper context, it reads like these trends should correspond to the level of OAE, but not the duration of each experiment.

*By specifying that the increase in pH and decrease in $pCO_2$ were persistent, we aimed to state that these conditions stayed relatively stable (the gradient remained) and different to ambient levels throughout the experiment. However, we explained this further by changing the sentence to: "These entailed a persistent increase in pH and decrease in $pCO_2$ when compared to ambient levels, since full natural equilibration throughout the duration of the experiment did not occur."*

22. Line 372-374, "addition" with quotation marks, I'd make it more explicit that the experimental technique used surrogate of chemical mixers instead of direct mineral additions.

*Instead of addition in quotation marks, we stated that we undertook addition simulations and specified "with compounds containing..." the key elements present in the two minerals, in parentheses: "Especially in terms of community composition with calcium-based OAE treatments through the hydrated lime addition simulation (with compounds containing $Ca^{2+}$ and $OH^-$ separately), potentially increasing the abundance of pelagic calcifiers, and silicate-based OAE through the forsterite one (with compounds containing $Mg^{2+}$, $SiO_3^{2-}$, and $OH^-$ independently), favoring diatom proliferation."*

23. Line 409, this 1% fraction of DIC as CO2 (which should be aqueous CO2) is salinity and temperature-dependent, so some context is needed.

*True. We added "~''1% to specify that this is an approximation and stated that we are referring to the DIC in seawater.*

24. Line 415-421, this discussion needs to be placed in the context of the study region to make it the case.

*In parentheses we reiterated that Ruanefjorden is our study site's location.*

25. Line 435, the ratio of observed Si and N uptake hinges upon the nutrient handling methods. Hence the ratio needs to be taken with a grain of salt.

*The inorganic nutrient determination methods that are in question have been used over 1200 and 1400 times since 1999. Additionally, in the article put forward (Reed, et al., 2023) they evaluate how higher pore sizes and longer times until analysis reduce the PO4 and NH4*

*determination reliability, particularly when baseline nutrient concentrations are very low. In this part of the discussion section on nutrient uptake, we focus on Si: N ratios, in which N is derived from NO3 uptake (not NH4, nor NOx), during phase II. Thus, after a significant nutrient addition.*

26. Line 451, "in terms of", meaning?

    *Here "in terms of growth rates" was removed.*

27. Line 455, clarify what the statement means.

    *Thank you for noticing. This sentence was re-written as follows: "This would therefore explain why, when GP is normalized to Chla (assimilation numbers), no differences between mineral treatments can be inferred, even if they occurred, in absolute GP:Chla temporal development, nor in their overtime response to the ΔTA gradient (Figure 7)."*

28. Line 465, remove "herein".

    *Removed.*

29. Line 467, the fact that Daphnia is a zooplankton needs to be mentioned here.

    *Specified in line 474 ("…Daphnia, the keystone herbivorous zooplankton species…").*

30. Line 484, "stronger" should be replaced with something like higher levels of chemical modification of seawater.

    *Addressed by changing "stronger OAE deployments" to "higher TA levels"*

The supplemental materials could use more help with higher resolution figures.

*Addressed. We want to thank the reviewer for noticing.*

**References**

Bach, L. T., Gill, S. J., Rickaby, R. E. M., Gore, S., & Renforth, P. (2019). CO2 Removal With Enhanced Weathering and Ocean Alkalinity Enhancement: Potential Risks and Co-benefits for Marine Pelagic Ecosystems. *Frontiers in Climate*, *1*(October). https://doi.org/10.3389/fclim.2019.00007

Foster, G. L., Pogge Von Strandmann, P. A. E., & Rae, J. W. B. (2010). Boron and magnesium isotopic composition of seawater. *Geochemistry, Geophysics, Geosystems*, *11*(8), 1–10. https://doi.org/10.1029/2010GC003201

Hansen, H. P., & Koroleff, F. (1999). Determination of nutrients. *Methods of Seawater Analysis*, 159–228.

Hauck, J., Köhler, P., Wolf-Gladrow, D., & Völker, C. (2016). Iron fertilisation and century-scale effects of open ocean dissolution of olivine in a simulated CO2 removal experiment. *Environmental Research Letters*, *11*(2). https://doi.org/10.1088/1748-9326/11/2/024007

Holmes, R. M., Aminot, A., Kérouel, R., Hooker, B. A., & Peterson, B. J. (1999). A simple and precise method for measuring ammonium in marine and freshwater ecosystems. *Canadian Journal of Fisheries and Aquatic Sciences*, *56*(10), 1801–1808. https://doi.org/10.1139/f99-128

Köhler, P., Abrams, J. F., Völker, C., Hauck, J., & Wolf-Gladrow, D. A. (2013). Geoengineering impact of open ocean dissolution of olivine on atmospheric CO2, surface ocean pH and marine biology.

*Environmental Research Letters*, *8*(1). https://doi.org/10.1088/1748-9326/8/1/014009

Renforth, P., & Henderson, G. (2017). Assessing ocean alkalinity for carbon sequestration. *Reviews of Geophysics*, *55*(3), 636–674. https://doi.org/10.1002/2016RG000533

---

## Author Comment (AC3)

**RC2 specific comments:**

This manuscript investigated the microbial responses to OAE approaches based on mesocosm experiments. The data presented are valuable in promoting the understanding of OAE impacts. My major suggestion for the authors is that, the discussion part could be expanded and polished a bit more, so that readers can get some take-home messages more easily. Right now, all figures are in the Results section, and they are all time-evolution of measured values. It is hard to extract key points from these figures. More in-depth analysis of these data would be helpful.

*We would like to thank the reviewer for taking the time to offer these suggestions. We have addressed the comment on the discussion section by expanding it and by dividing the original section "4.1 Responses to the carbonate chemistry conditions and mineral type", and its content, into two separate sections: "4.1 Response to the carbonate chemistry conditions: Non-equilibrated TA gradient" and "4.2 Response to the two simulated mineral additions: Calcium- vs silicate-based." We hope this restructuring provides clearer take-home messages for the readers.*

*Regarding the statistical analysis, it is challenging to statistically detect the signal of the response, partially due to the experimental setup (comparing 5 treatments against 5) and partly because the observed effects are quite mild. For instance, there is no noticeable effect of the TA gradient (and the associated changes to $pCO_2$ and pH) on the absolute values of any of the parameters included in this study. In fact, we conducted ANCOVAs, amongst other tests, to determine if there was a statistically significant effect of the mineral type, of the TA gradient, or of the combination of both, during separate phases within phase II, using averages. However, the data did not meet the assumptions of normal distribution or homogeneity of variances, even after transformation. Thus, even if significance was found in some cases, these analyses were not included.*

*Given these challenges, and the main take-home message (the found delay in the measured production rates and the chlorophyll a concentration after the nutrient addition, and in response to the TA gradient), we plotted the slopes of daily linear models over time, alongside presenting time-evolution figures of measured values. This approach illustrates that the low TA treatments responded sooner than the high TA ones. We have also addressed the comment on statistics by providing the plots of all the daily linear models employed to produce the mentioned figures as supplementary materials.*

Another concern is that the manuscript seems quite colloquial. For example, Line 378-381, "The experiment was started under post-bloom conditions (Figure 2). Nutrient concentrations were low when the treatments were applied. Thus, an initial response in the microbial community production and respiration rates to the TA manipulation was likely concealed by the nutrient limitation. Actually, after a mixing event was simulated on day 26 and 28, a response could be discerned."

Also, Line 386, "In the past, prior to the emergence of ocean acidification as a focal point in scientific inquiry…", Line 459-460, "This difference between the controls was unexpected. This was probably a random response caused by the mesocosm effect…" etc. These do not sound like scientific languages, and I would recommend that the authors revise their expressions throughout the manuscript.

*We regret that our manuscript seemed colloquially written. We have addressed these comments by re-writing the listed sentences as follows:*

*Line 378-381: The experiment commenced under low inorganic nutrient concentrations (Figure 2), characteristic of post-bloom conditions. Thus, an initial response in the microbial community production and respiration rates to the TA manipulation was likely concealed by the nutrient limitation.*

*Following a simulated mixing event on days 26 and 28, a discernible response emerged. The results from the current study show a delay in bloom formation with increasing TA, when such manipulation is non-equilibrated.*

*Line 386: In the past, before ocean acidification emerged as a central focus of scientific research…*

*Line 459-490: The observed discrepancy between the controls was unanticipated and may have resulted from a random response attributable to the mesocosm effect.*

*This type of correction has also been applied, to the best of our abilities, throughout the whole manuscript.*

Some minor comments:

Figure 1, I don't see the Fjord data but it is in the legend. I think the authors was hoping to add the legend to Figure 2, as Figure 2 has the data, but the legend does not have 'Fjord'.

*We want to thank the reviewer for noticing. The legends have been exchanged as suggested.*

Line 358-360, "The latter are both being…" needs to be corrected.

*This comment has been addressed by re-writing the start of the first paragraph of the discussion as follows:*

*"The experiment described herein involved the application of two distinct alkalinity gradients utilizing two ocean alkalinity enhancement (OAE) strategies based on different minerals: hydrated lime and forsterite. Forsterite is the Mg endmember of olivine, which occurs commonly in nature, while hydrated lime is produced through the calcination of limestone* (Renforth and Henderson, 2017). *Both minerals have been considered for OAE implementation; however, experimental studies on their potential impacts on natural communities remain unaddressed."*

---

## Author Comment (AC4)

**RC3 Specific comments:**

This study explores non-equilibrated Ocean Alkalinity Enhancement (OAE) using silicate and calcium-based Total Alkalinity (TA) gradients (0 to 600 µmol · L-1) under natural conditions. The manipulation increased pH and decreased pCO2, impacting bloom formation after macro-nutrients were added. Overall, this study contributes to the current understanding of OAE field application. Here are my comments to help the authors refine this manuscript.

Major comments:

1. The authors proposed that the addition of TA higher than 150umol/L had delayed the bloom of phytoplankton, but the results may not be strong evidence for this argument. The GP-ΔTA, Chl-a-ΔTA, and CR-ΔTA etc were analysed as shown in Figs.4,5,6 with a CI bar representing the Confidence Interval (CI). If I understand correctly, if CI bars overlap with the horizontal line (y=0), the GP or Chla and other parameters don't have a significant linear relationship with the ΔTA on that day. Considering the CI bars in many of the subplots in Figs. 4,5,6 are large (especially during the bloom time), so this argument about delayed bloom may be overinterpreted. In addition, in Fig.4 A and C, the peak of the curve seemed to occur earlier in the Si 150 treatment than in the Si 0 treatment, which conflicts with the argument. I would suggest the authors reconsider this argument throughout the manuscript.

*We appreciate the reviewer's critical feedback and acknowledge the potential overinterpretation of our results, particularly when considering the Si control. This specific mesocosm displayed an atypical behaviour, differing notably even from the Ca control. It exhibited two small peaks in gross production (GP), net community production (NCP), and chlorophyll a (Chla), with a delayed response compared to the Si 150 treatment. We have discussed this anomaly in the discussion section, specifically on line 456 of the initial preprint, where we mentioned this may have been due to the mesocosm effect. In fact, it is worth noting that the average contribution during phase I of CR to GP in the silicate control was the highest at 63%. Thus, right before the nutrient addition that led to the described pattern, the community was likely much different in that treatment than the rest*

*The CI bars show that statistical significance was not found when these cross the y axis and, in most cases, are quite wide. This is mainly because of the variability observed on each individual day. The daily linear models employed to produce the effect size graphs have been included in the supplementary materials. As can be observed, the lower treatments do respond slightly earlier than the high TA ones, generating negative slopes that later invert, although most are not significant. We argue that this is because of the small sample size (n = 5 in the Si and Ca gradients separately). In fact, the variability that led to non-significant negative daily linear trends in terms of GP and Chla was due to just two treatments: The control in the silicate treatment set, and the ΔTA 150 in the calcium one (which experienced a longer delay than the control).*

*While we acknowledge that these results are qualitative and not statistically significant (which has been highlighted further across the entire manuscript, and a paragraph addressing this has been added at the end of the results section), we believe they are still noteworthy. In fact, the delayed bloom formation in the high TA treatments has subsequently been observed in two follow-up mesocosm studies conducted a posterior (unpublished). Further, this trend, though subtle, emerged even after a nearly month-long acclimation period. Thus, the fact that when*

*closing the mesocosms, increased variability in community composition and structure amongst them is prompted, we believe further reinforces our argument. Additionally, we do not claim the delay to be proportional to the TA addition, but rather described it as a mild lag with lower TA treatments responding sooner. Despite being a qualitative observation in this specific experiment, we consider this a key result that warrants further study due to its potential implications.*

2. The design of macronutrient fertilization is a good way to understand the field application in other seasons, but there is limited information about why authors choose these certain macronutrient levels. Is it close to the real nutrient levels in different seasons? Please explain more details about the design in lines 251-260.

*We appreciate the reviewer's insightful comment. To address this, we have expanded the methods section "2.2 Carbonate chemistry manipulation and nutrient fertilization" to clarify our rationale behind the chosen macronutrient levels. The amount of $NO_3$ of 4 μmol/L was added to simulate upwelling of deeper nutrient rich waters, creating a phytoplankton bloom comparable in biomass to natural occurrences in the area. The N:P ratio was targeted at 16:1, aligning with the Redfield ratio, which represents the standard nutrient composition of marine phytoplankton and is widely recognized in ecological studies. The Si:N ratio was set at approximately 1:4. This design choice was made to create a niche for coccolithophores so that they would not be outcompeted by diatoms, which would be silicate limited under these conditions* (Gilpin et al., 2004; Schulz et al., 2017).

3. In discussion 4.1, I appreciate the authors trying to explain the reasons why a potential delayed bloom would occur. However, there is no sufficient information about local phytoplankton community composition in the experimental sites. Were there diatoms? Were there calcifying phytoplankton? In Fig.3, the peak of Chla was similar or even higher than the Fjord, is it possible some species benefited from the addition of TA? Considering the pH tolerance and CO2 utilization are species-specific, the information about local phytoplankton will be useful.

*Data on phytoplankton community composition and structure were not made available for this publication since its focus was on the trends observed in terms of microbial metabolic rates. A separate publication that is currently in preparation will report and discuss the observed changes to the community composition in relation to the trends detected here and the carbonate chemistry conditions.*

*There were diatoms present* (Ferderer et al., 2023). *Although indeed, information about the local phytoplankton community would be very useful (added to the manuscript at the end of the third paragraph in section 4.1). For now, plausible hypotheses, based on the relationship of pH and pCO2 with growth rates being species-specific, are presented.*

*The difference between the fjord and the mesocosms is interesting. The phytoplanktonic communities were definitely different considering their GP:Chla, as well as the differences in Chla concentrations described by the reviewer. Leading to the conclusion that certain species benefited from the carbonate chemistry conditions inside the mesocosms. Although the mesocosm effect may have also played a role in this differentiation, this has been added at the end of the section 4.1's 4th paragraph, as well as in the results section.*

Minor comments:

1. Line 76 "iron, which is a co-limiting micronutrient": please add a reference.

   *Addressed*

2. Line 130: five mesocosms?

   *Addressed*

3. Line 162: please add a reference about how you correct the pH.

   *Addressed*

4. Line 200: Where is the T in the equation?

   *Addressed*

5. Line 230: The unit of TA is umol/L in previous paragraphs, and the delta TA unit is uEq/L in table 1. Please double-check and explain uEq/L.

   *Addressed. We want to thank the reviewer for noticing. They were meant to be all in umol/L.*

6. Line 236: "The difference between … was quite steep", please rephrase this sentence.

   *Addressed*

7. Line 265: "In the silicate ones, …" should be "calcium ones"?.

   *"In the silicate ones…" here is correct. Initially, the average depletion of $Si(OH)_4$, $NO_3^-$, and $PO_4^{-3}$ in the calcium treatments are listed. Afterwards, the same is described for the silicate treatments. First the consumption of $NO_3^-$, and $PO_4^{-3}$ in the latter, which were alike to those observed in the calcium treatments. The $Si(OH)_4$ consumption in the silicate treatments is stated afterwards separately because of the different concentrations of this nutrient between the two sets of treatments throughout.*

8. Line 268-270: It looks like the drawdown of N and P in Ca treatments was just as much as in Si treatments. Therefore, they were both N and P limited.

   *We agree with the reviewer and this comment has been addressed by removing this sentence altogether.*

9. Line 274: What's the grey line? Please explain the greyline, the dash lines, the phases I and II here.

   *Addressed in all figure captions*

10. Line 297: Please explain more about the top subplots in Fig. 4 and Fig. 5. Does the positive value mean the positive relationship between the parameter and delta TA?

    *Indeed, if the slope is positive, it means there may be a positive (although not necessarily significant) relationship between the parameter in question and the delta TA. The statistical strength is though lacking because of the small sample size (as explained above). These plots are just a representation of a trend that, with this experimental setup, cannot be statistically "verified". This has been further clarified in a paragraph added at the end of the results section and in the conclusions.*

11. Line 308-315: Please consider moving this paragraph to the method.

*We want to thank the reviewer for this suggestion. We believe however that, since these correlations were carried out once the data were obtained and processed, that it belongs in the results section.*

12. Line 358: Please explain more about what hydrated lime and forsterite are.

    *Addressed*

13. Line 374 -376: Please add references.

    *Addressed*

14. Line 399 "There was high variability in their tolerance to high pH": what is the "high pH" range?

    *Addressed.* Hansen (2002) *reports some species stopped growing at pH 8.3-8.4, while other were still able to grow at pH 10. Thus, the range stipulated here: 8.4 to 10.*

15. Line 415-419: Please add references.

    *Addressed*

16. Line 436: "In the latter, the community", please state the specific phase or day.

    *This sentence has been removed upon realizing that it was not pertinent to the argument being presented.*

17. Line 441-443: The relationship between diatom silicifying and Chl-a concentration is not clearly explained here. Please rephrase the sentence.

    *This sentence has been extended and modified: "Diatoms in these treatments may thus have been able to allocate more resources to growth and photosynthesis (Inomura et al., 2023). Therefore, the excess availability of silicate could have aided the faster response to the mixing event simulation, as well as contributed to the higher absolute GP and Chla concentrations observed during phase II in the silicate treatments, compared to the calcium ones".*

**References**

Ferderer, A., Schulz, K. G., Riebesell, U., Baker, K. G., Chase, Z., and Bach, L. T.: Investigating the effect of silicate and calcium based ocean alkalinity enhancement on diatom silicification, Biogeosciences Discuss., 1–28, 2023.

Gilpin, L. C., Davidson, K., and Roberts, E.: The influence of changes in nitrogen: silicon ratios on diatom growth dynamics, J. Sea Res., 51, 21–35, https://doi.org/https://doi.org/10.1016/j.seares.2003.05.005, 2004.

Hansen, P. J.: Effect of high pH on the growth and survival of marine phytoplankton: implications for species succession, 28, 279–288, 2002.

Inomura, K., Pierella Karlusich, J. J., Dutkiewicz, S., Deutsch, C., Harrison, P. J., and Bowler, C.: High Growth Rate of Diatoms Explained by Reduced Carbon Requirement and Low Energy Cost of Silica Deposition, Microbiol. Spectr., 11, https://doi.org/10.1128/spectrum.03311-22, 2023.

Schulz, K. G., Bach, L. T., Bellerby, R. G. J., Bermúdez, R., Büdenbender, J., Boxhammer, T., Czerny, J., Engel, A., Ludwig, A., Meyerhöfer, M., Larsen, A., Paul, A. J., Sswat, M., and Riebesell, U.: Phytoplankton blooms at increasing levels of atmospheric carbon dioxide: Experimental evidence for negative effects on prymnesiophytes and positive on small picoeukaryotes, Front. Mar. Sci., 4,

https://doi.org/10.3389/fmars.2017.00064, 2017.

---

## Author Response (AR2)

Dear Dr. Middleburg,

We greatly appreciate your time and feedback. We have carefully addressed both comments provided by the referee as follows:

- We have added the sentence, "The targeted and established carbonate chemistry treatments are compared here." at the beginning of Table 1's caption.

- The information about the intervals between sample collection and filtration, and between filtration and analysis, has been added to the methods section on inorganic nutrient analysis. We apologize for the oversight.

We look forward to your response.

Kind regards,
Laura Marín-Samper, and co-authors